# Using positional information to provide context for biological image analysis with MorphoGraphX 2.0

**Sören Strauss[1], Adam Runions[1†], Brendan Lane[1,2], Dennis Eschweiler[3], Namrata Bajpai[1], Nicola Trozzi[1,2], Anne-Lise Routier-Kierzkowska[4], Saiko Yoshida[1], Sylvia Rodrigues da Silveira[4], Athul Vijayan[5], Rachele Tofanelli[5], Mateusz Majda[1,2], Emillie Echevin[4], Constance Le Gloanec[4], Hana Bertrand-Rakusova[4], Milad Adibi[1], Kay Schneitz[5], George W Bassel[6], Daniel Kierzkowski[4], Johannes Stegmaier[3], Miltos Tsiantis[1], Richard S Smith[1,2]\***

[1]Max Planck Institute for Plant Breeding Research, Department of Comparative Development and Genetics, Cologne, Germany; [2]John Innes Centre, Norwich Research Park, Norwich, United Kingdom; [3]Institute of Imaging and Computer Vision, RWTH Aachen University, Aachen, Germany; [4]IRBV, Department of Biological Sciences, University of Montreal, Montreal, Canada; [5]Plant Developmental Biology, TUM School of Life Sciences, Technical University of Munich, Freising, Germany; [6]School of Life Sciences, University of Warwick, Coventry, United Kingdom

**\*For correspondence:**
Richard.Smith@jic.ac.uk

**Present address:** †Department of Computer Science, University of Calgary, Calgary, Canada

**Competing interest:** The authors declare that no competing interests exist.

**Abstract** Positional information is a central concept in developmental biology. In developing organs, positional information can be idealized as a local coordinate system that arises from morphogen gradients controlled by organizers at key locations. This offers a plausible mechanism for the integration of the molecular networks operating in individual cells into the spatially coordinated multicellular responses necessary for the organization of emergent forms. Understanding how positional cues guide morphogenesis requires the quantification of gene expression and growth dynamics in the context of their underlying coordinate systems. Here, we present recent advances in the MorphoGraphX software (Barbier de Reuille et al., 2015) that implement a generalized framework to annotate developing organs with local coordinate systems. These coordinate systems introduce an organ-centric spatial context to microscopy data, allowing gene expression and growth to be quantified and compared in the context of the positional information thought to control them.

## Editor's evaluation

Quantitative imaging has become a mainstay of modern cell and developmental biology. This article reports major advances in the image analysis software package MorphGraphX (MGX). MGX2.0 includes new tools for precise quantitation of cellular behaviors, such as cell division and expansion, within the context of positional information in the growing organs. This article is the go-to resource for current and future users of MGX to learn the power of the software package, with which they can quantify the spatiotemporal dynamics of the growth and development of living organisms.

## Introduction

Many aspects of animal morphogenesis are thought to be controlled by positional information (*Wolpert, 1969*), where cells can sense their position in a developing organ and respond accordingly. This phenomenon may be even more pervasive in plants as cells cannot relocate within organs

and must decide their fate based on their location. For example, root morphogenesis appears to be controlled by an organizing center at the root tip that provides founder cells and positional information to the growing structure (*Scheres et al., 2002*). Ablation of cortical cell initials in the root meristem causes the neighboring pericycle cells to divide and fill the available space, subsequently adopting the fate associated to their new location (*van den Berg et al., 1995*). A similar effect has been demonstrated for a variety of cell types in the *Arabidopsis* root (*Marhava et al., 2019*). In leaves, development is thought to be coordinated by polarity fields oriented from leaf base to tip (*Kierzkowski et al., 2019*; *Kuchen et al., 2012*). Over time organs can initiate new growth axes, such as when serrations or leaflets develop in more complex leaves (*Barkoulas et al., 2008*; *Kierzkowski et al., 2019*), or lateral roots emerge from the primary root (*Scheres et al., 2002*). In these cases, information from several organizers must be integrated to direct cell response.

To understand how positional information controls morphogenesis, it is necessary to quantify cell shape, gene expression, and morphogen concentration changes over time, preferably at the cellular level. This information then needs to be related to its position relative to the organizers controlling development within the organ. As computational power and imaging methods improve, new software packages for cell segmentation and lineage tracking are being developed (*Sommer et al., 2011*; *Stegmaier et al., 2016*), including many specialized for plants (*Barbier de Reuille et al., 2015*; *Eschweiler et al., 2019*; *Fernandez et al., 2010*; *Schmidt et al., 2014*; *Wolny et al., 2020*). This progress has enabled the segmentation of time-lapse data at increasingly higher resolution and throughput (*Hervieux et al., 2016*; *Kierzkowski et al., 2019*; *Sapala et al., 2018*; *Willis et al., 2016*). Although this increase in data volume offers tremendous potential to understand how genes control form, the analysis of geometric data from thousands of cells is nontrivial. Information about a cell's shape, gene expression, and growth directions is of limited value when the cell's spatial context within the developing organ is unknown.

MorphoGraphX is a computer software platform that is specialized for image processing on surface layers of cells (*Barbier de Reuille et al., 2015*). It has proven especially useful for the analysis of confocal microscopy images from time-lapse data in order to quantify the cellular-level dynamics of growth, cell division, and gene expression (e.g., *Bringmann and Bergmann, 2017*; *Feng et al., 2018*; *Hervieux et al., 2016*; *Hong et al., 2016*; *Kierzkowski et al., 2019*; *Louveaux et al., 2016*; *Sapala et al., 2018*; *Scheuring et al., 2016*; *Tsugawa et al., 2017*; *Vlad et al., 2014*; *Zhang et al., 2020*; *Zhu et al., 2020*; *Fridman et al., 2021*). Key to the approach taken in the software is the representation of cell layers as curved, triangulated surface meshes that capture the overall 3D shape of organs, which retains much of the simplicity of 2D segmentation and lineage tracking. These '2.5D' images contain the geometry of the sample at two scales. The global shape of the organ is captured by the mesh's geometry, while a cellular-scale representation is obtained from the confocal signal projected onto the mesh, which is segmented to extract the shape of individual cells on the surface (*Figure 1A–C*). When combined with time-lapse data acquisition and cell lineage tracking, MorphoGraphX allows cell growth and its relationship to gene expression to be quantified (*Figure 1D and E*; *Kierzkowski et al., 2019*; *Sapala et al., 2018*; *Vlad et al., 2014*). In addition to cell surface analysis, MorphoGraphX also supports the creation and analysis of full 3D meshes with volumetric cells (*Figure 1—figure supplement 1*; *Vijayan et al., 2021*). Here, we describe new methods we have developed in MorphoGraphX to understand these data by additionally annotating cells with positional information. Not unlike the annotation of sequence data, this allows cellular data to be given spatial context, and a frame of reference within the organ relative to its developmental axes and the organizers instructing morphogenesis.

## Results and discussion

Most workflows in MorphoGraphX begin by converting 3D image stacks into meshes of 2.5D or 3D cellular segmentations, which are created directly on voxels in the case of 3D segmentation, or on surface meshes in the case of 2.5D data (*Figure 1A–C*, *Figure 1—figure supplement 1A and B*). Recent advances in voxel classification with convolutional neural networks (CNNs) for cell boundary prediction can improve input images and the resulting segmentation (*Eschweiler et al., 2019*; *Eschweiler et al., 2021*; *Wolny et al., 2020*), especially for 3D segmentations. Although a selection of these and other image denoising and preprocessing tools are available directly within MorphoGraphX, it is also possible to preprocess and/or segment 3D images in other software and import them into

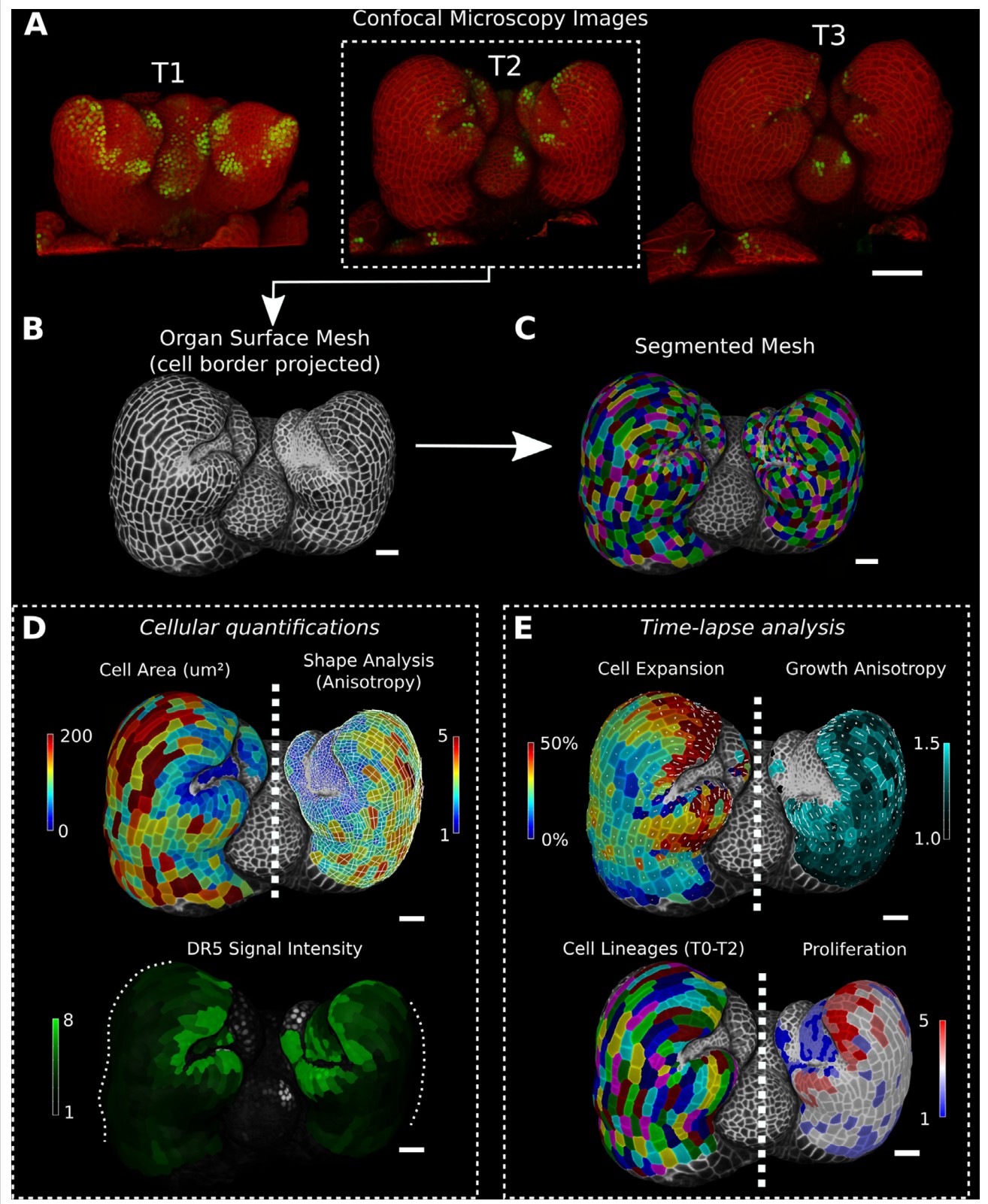

**Figure 1.** Cellular segmentation and basic quantifications supported by MorphoGraphX demonstrated by using a time-lapse series of an *A. thaliana* flower meristem. (**A**) Multichannel confocal microscopy images with a cell wall signal (red) and DR5 marker signal (green). Shown are the last three time points (T1–T3) of a four-image series (T0–T3). (**B, C**) Extracted surface mesh of T2. Cell wall signal near the surface was projected onto the curved mesh to enable the creation of the cellular segmentation in (**C**). The segmented meshes provide the base for further analysis within MorphoGraphX as shown

*Figure 1 continued on next page*

*Figure 1 continued*

in (**D**) and (**E**). (**D**) Top: MorphoGraphX allows the quantification of cellular properties such as cell area and shape anisotropy (shown as heat maps). The white axes show the max and min axes of the cells. Bottom: heat map of the quantification of the DR5 marker signal (arbitrary units) projected onto the cell surface mesh. (**E**) When cell lineages are known, time-lapse data can be analyzed. Top: heat maps of cell area expansion and growth anisotropy (computed from T1 to T2). The white crosses inside the cells depict the principal directions of growth. Bottom: visualization of the cell lineages and heat map of cellular proliferation (number of daughter cells), computed from T0 to T2. Scale bars: (**A**) 50 μm; (**B– E**) 20 μm. See also user guide Chapters 1–15 and tutorial videos S1 and S2 videos S1 and S2available at https://doi.org/10.5061/dryad.m905qfv1r.

The online version of this article includes the following figure supplement(s) for figure 1:

**Figure supplement 1.** Basic 3D analysis using MorphoGraphX demonstrated using an *Arabidopsis* ovule.

MorphoGraphX for further analysis. After the initial segmentation, cellular features can be quantified, such as cell area, shape, and gene expression for single time points, or cell proliferation and growth for time-lapse data (*Figure 1D and E*). These data can then be annotated with positional information to aid the understanding of the development of the organism under study. Even with deep learning techniques, image quality needs to be very high for full 3D segmentation, and this is often not possible with live imaged data. By enabling image processing on 2.5D surface images, Morpho-GraphX can be used in many systems on live imaged data where full 3D segmentation is currently not possible (*Hervieux et al., 2016*; *Kierzkowski et al., 2019*; *Sapala et al., 2018*; *Silveira et al., 2022*; *Vlad et al., 2014*). Annotation with coordinate systems can further reduce image quality requirements in cases where growth along a single dimension is required (*Liu et al., 2022*).

## Defining coordinates within an organ

The simplest method to provide positional information for the cells in a sample is by aligning the sample with a set of 3D coordinate axes (*Figure 2A*). For example, a developing root meristem can be aligned and positioned such that the organizing quiescent center is at the origin with the Y-axis increasing in the longitudinal direction of the root. Provided the sample is reasonably straight, this allows cellular measures to be compared with their distance from the quiescent center (*Figure 2B*).

However, for curved organs significant errors will occur, especially in more distal regions, further from the origin. In this case, the central axis can be defined by a curved line that conforms to the curvature of the organ (*Figure 2C*; *Schmidt et al., 2014*; *Montenegro-Johnson et al., 2015*). In MorphoGraphX, this line can be represented by a Bezier spline (*Bézier, 1968*) with control points positioned using either interactive manipulation or automatically from a selected file of cells. Distance can then be calculated along the line and transferred to cells in the cross section perpendicular to the line (*Figure 2C*). MorphoGraphX also allows a 2D Bezier surface to be positioned next to or within a sample, enabling two directions to be aligned with the natural curvature of the sample.

Placing the Bezier curve or surface to curved organs with more complex shape is challenging. An alternative method is to select one or more cells at a reference position and calculate the distance relative to the selection (*Figure 2D*, *Video 1*). This offers an easy method to create a distance field and greatly increases the variety of organs that can be accommodated. The distance is determined by computing the shortest path along cells through the tissue, causing it to naturally follow the curvature of the organ.

Once cells have been annotated with positional information, it can be used for the analysis of cell-level data, such as growth, cell proliferation, cell shape, and gene expression. Using the distance measure to define the proximal-distal (PD) axis on an *Arabidopsis* sepal (*Figure 2D*), geometric measures can be plotted against the local coordinate system. In *Figure 2F*, cell area extension was plotted against distance from the base of the sepal. On the full 7-day sepal time-lapse shown in *Figure 2—figure supplement 1* (*Hervieux et al., 2016*), initially growth is predominantly located at the distal parts of the organ, followed by the progressive displacement of the high growth zone towards the base of the sepal (*Figure 2—figure supplement 1A*). By time point 6, the growth has slowed and become more uniform as the organ differentiates. Proliferation is initially more uniform, but otherwise follows a pattern similar to growth, progressing basally as the organ matures (*Figure 2—figure supplement 1B*). The data can be indexed by position from the base of the sepal and plotted, showing how growth and proliferation vary along the PD developmental axis as a percentage of the total sepal length (*Figure 2—figure supplement 1D and E*). It can be seen in the graphs that although the sepal appears to undergo a similar growth arrest starting at the tip as the *Arabidopsis*

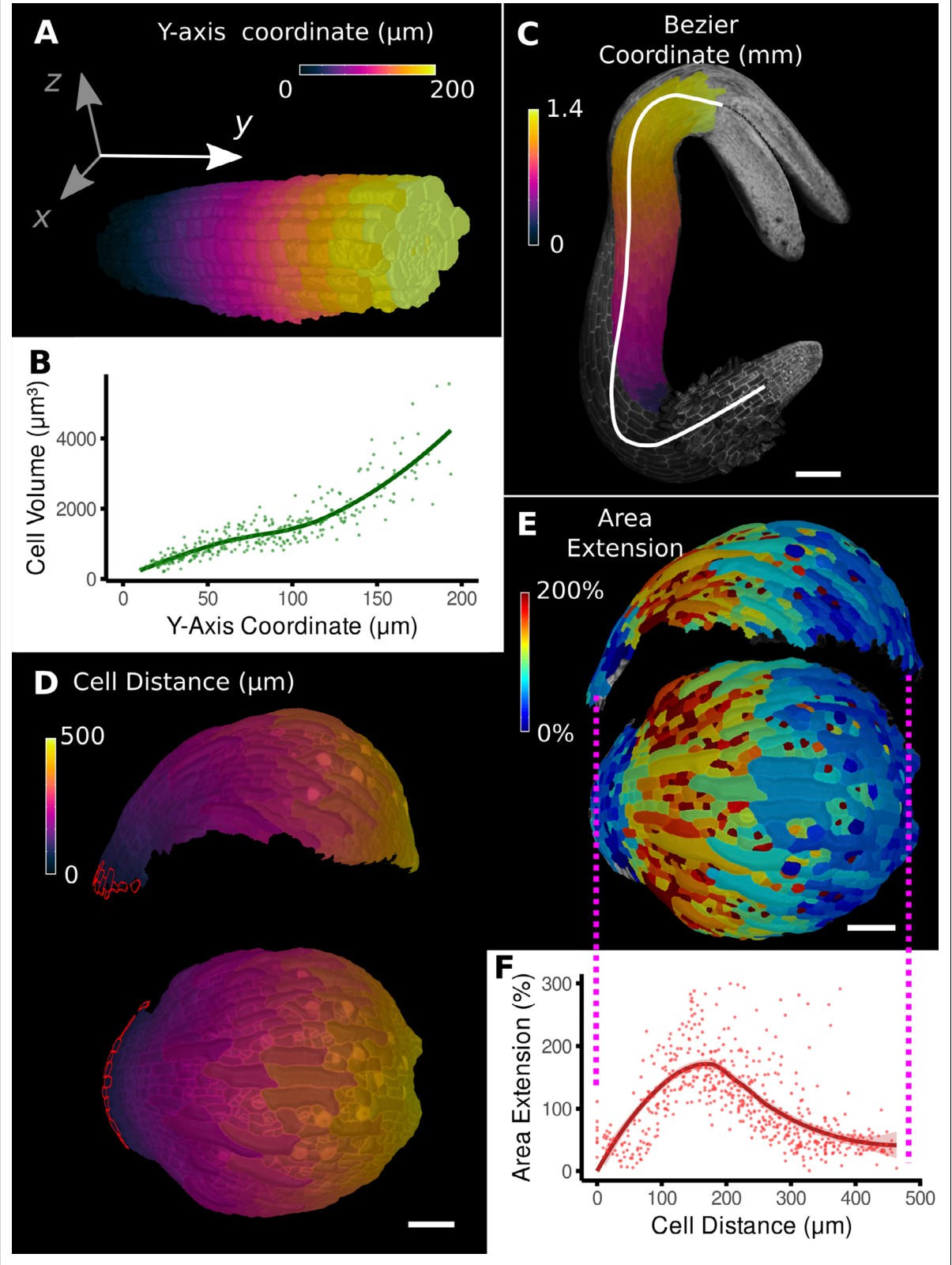

**Figure 2.** Methods to define positional information and their application to data analysis in plant organs. (**A**) Y-axis aligned *A. thaliana* root. The cells are colored according to the y-coordinate of their centroid position. (**B**) Plot of cell volumes of epidermis cells of the root in (**A**) along the y-axis with a fitted trend line. (**C**) Seedling of *A. thaliana* with a surface segmentation of the epidermis. A manually defined Bezier curve (white) allows the assignment of accurate cell coordinates along a curved organ axis. (**D**) Side and top views of an *A. thaliana* sepal with a proximal-distal (PD) axis heat coloring. The

*Figure 2 continued on next page*

*Figure 2 continued*

cell coordinates were assigned by computing the distance to manually selected cells (outlined in red) at the organ base. This method allows organ coordinates to be assigned in highly curved tissues. (**E**) Side and top views of (**D**) with a heat map coloring based on cellular growth to the next time point. (**F**) Plot summarizing the growth data of (**E**) using the PD-axis coordinates from (**D**). See *Figure 2—figure supplement 1* for the analysis of the complete time-lapse series. Scale bars: (**A**) 20 µm; (**C**) 100 µm; (**D, E**) 50 µm. See also user guide Chapter 23 'Organ-centric coordinate systems' and tutorial video S3video S3 available at https://doi.org/10.5061/dryad.m905qfv1r.

The online version of this article includes the following figure supplement(s) for figure 2:

**Figure supplement 1.** From cellular resolution heat maps to a global analysis of *A. thaliana* sepal development using organ-centric coordinates.

leaf, there are subtle differences. The growth in the early stages is more distal in the sepal, along with the proliferation, and the zone of higher growth moves towards the base as the sepal develops. This is in contrast to the *Arabidopsis* leaf where the growth and proliferation zones remain relatively fixed with respect to the total leaf length (*Figure 1K and M*; *Kierzkowski et al., 2019*). The use of relative coordinates also makes it possible to pool data from multiple samples (*Vijayan et al., 2021*; *Zhang et al., 2020*) and compare data from different genotypes (*Kierzkowski et al., 2019*; *Montenegro-Johnson et al., 2019*; *Zhang et al., 2020*).

## Deriving directions from organ coordinates

In addition to scalar information such as areal growth rate or cell volume, MorphoGraphX can also quantify directional information, such as the principal directions of growth (PDGs) that represent the maximal and minimal directions of growth for each cell (*Figure 3A–C*). A common problem with the interpretation of such directional information is the tendency for directions to be locally heterogeneous when growth is nearly isotropic. This happens because the maximal and minimal growth amounts are almost the same, and the displayed directions become arbitrary and heavily influenced by noise. This can make the comparison of growth directions between neighboring cells difficult. A more informative approach is to look at growth with respect to the directions of the developmental axes. This can be done by first setting up an axis defining the positional information for the leaf, for example, by using the previously introduced distance field (*Figure 2D*, *Figure 3B*). The growth directions are then projected onto this developmental axis and separated into components that are parallel (*Figure 3D*) and perpendicular (*Figure 3E*) to the axis. Using this method on the *Arabidopsis thaliana* leaf primordium different developmental zones with varying growth rates along the PD and medial-lateral (ML) axis can be revealed (*Figure 3C–E*): while the area extension differs greatly in midrib/petiole (low) and leaf blade cells (high), it can be seen that those differences mainly follow from the ML growth rate, whereas the PD growth map is more similar in these domains. Moreover, an increase in ML growth around the forming serration can be seen, separating it from the surrounding leaf blade cells that show less growth along this direction (*Figure 3E*). From the original PDG visualization, it is not immediately apparent that the varying ML growth is the main cause of the differences in the domains (*Figure 3C*).

Another benefit of looking at PDGs in the context of a local coordinate system is that it can provide a more direct comparison to the outputs of computational simulations. Developmental models of emergent organ shape often use morphogens that are thought to specifically control growth in parallel and perpendicular to a developmental axis (*Kierzkowski et al., 2019*; *Kuchen et al., 2012*; *Whitewoods et al., 2020*). By projecting the PDGs onto this axis, it is possible to directly compare model growth rates in the different directions to experiments. Since MorphoGraphX can load a wide variety of mesh

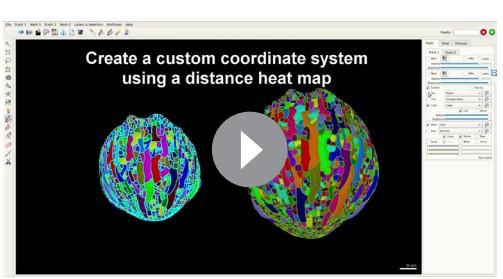

**Video 1.** Creation of a simple organ-centric coordinate system and its application on cellular data. Starting with two time points of an *Arabidopsis* sepal that have been lineage traced, the principal directions of growth are computed. Next, cells at the base of the sepal are selected to create a distance field that is used as a simple coordinate system. The growth (areal extension) can then be computed along the directions defined along the organ-centric coordinate system. The organ coordinates can also be exported and used to plot different kinds of cellular data.

https://elifesciences.org/articles/72601/figures#video1

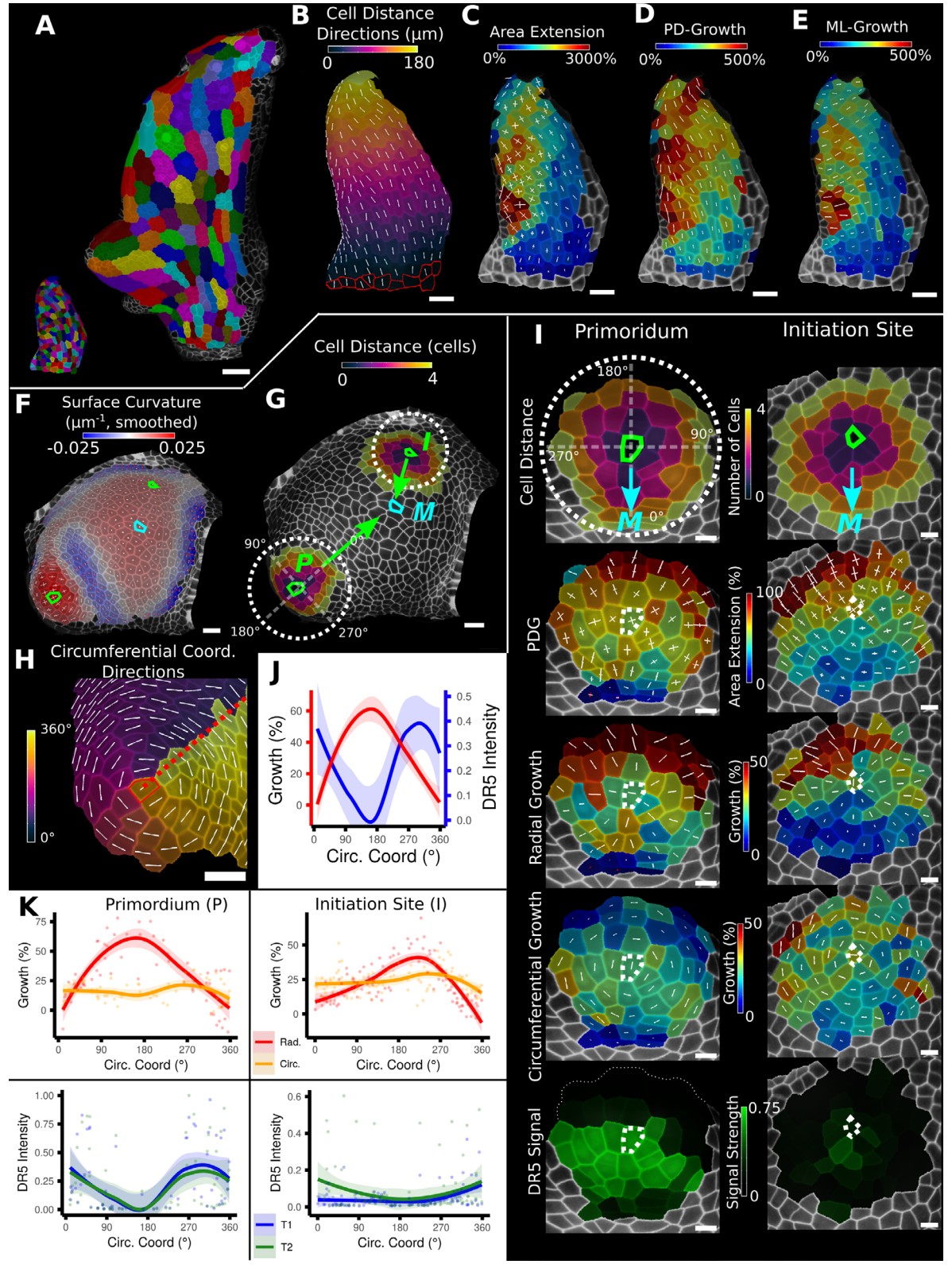

**Figure 3.** Examples of data analyses using organ coordinate directions. (**A–E**) Quantification of cellular growth along organ axes in a young *A. thaliana* leaf. (**A**) Segmented meshes of the leaf primordium at 3 and 6 days after initiation shown with cell labels and lineages of the earlier time point (3 days). (**B**) Earlier time point of (**A**) with proximal-distal (PD) axis coordinates (heat map) and directions (white lines) computed from selected cells at the leaf base. (**C**) Area extension (heat map) and principal directions of growth (PDGs, white lines) between the time points of (**A**). PDG axes are computed per

*Figure 3 continued on next page*

*Figure 3 continued*

cell and can point in different directions. (**D, E**) Computation of the growth component of (**C**) that is directed along the PD and the orthogonal medial-lateral (ML) axis. (**F–K**) Quantification of locally directed growth in leaf primordium and initiation site of a tomato meristem. (**F**) Smoothed heat map of cell curvature. Local maxima in this heat map (green and cyan cells) were selected as meristem center (M), primordium center (P), and initiation site (I) as shown in (**G**). (**H**) To analyze the data, we defined circumferential coordinate systems with their axes directions (white lines) around the primordium and initiation center (not shown), and aligned them towards the meristem center. (**I**) Heat maps of cell distance, area extension, radial and circumferential growth, and normalized DR5 signal intensity of the aligned primordium and initiation site. (**J**) Plotting the data of (**I**) reveals a negative correlation of the DR5 signal intensity and radial growth around the developing primordium. (**K**) Detailed plots of radial (red) and circumferential growth (orange) as well as the normalized DR5 signal intensity of the primordium and initiation site. Scale bars: (**A**) 50 µm; (**B–H**) 20 µm; (**I**) 10 µm. See also user guide Chapters 16 'Custom axis directions,' 23 'Organ-centric coordinate systems,' and tutorial video S3 available at https://doi.org/10.5061/dryad.m905qfv1r.

formats, this allows the direct comparison of similar quantifications made on templates extracted from model simulations from various sources.

*Figure 3F–K* shows a similar growth quantification for the tomato meristem (*Kierzkowski et al., 2012*), where local organ coordinates were created by using cell distance measures around each emerging leaf primordium with directions pointing towards (radial) and around (circumferential) their respective center. In addition to growth, the signal intensity of the auxin reporter DR5 was quantified in the same sample, allowing a direct comparison to auxin signaling levels and cellular growth. For both primordia, we found radial growth to have a high negative correlation with DR5 signal intensity. *Figure 3J* shows that the radial growth (red) peaks on the abaxial side of the emerging primordium, whereas the DR5 signal (blue) is higher on the adaxial side. Circumferential growth was more or less constant. The DR5 maximum tends to be on the adaxial side of the initiating leaf, whereas growth is much higher on the opposing abaxial side. This supports the idea that auxin acts as a trigger for primordium initiation (*Reinhardt et al., 2003*; *Smith et al., 2006*) rather than via controlling growth rates directly.

## Combining directions

In 2D or on 2.5D surfaces, local directions can be fully defined by a single distance measure by taking one direction aligned with the gradient of the distance field or a Bezier curve, and the other perpendicular to the first. This is similar to the methods used to specify directions in developmental modeling in plants (*Green et al., 2010*; *Kennaway and Coen, 2019*; *Kierzkowski et al., 2019*; *Kuchen et al., 2012*; *Whitewoods et al., 2020*), and thus facilitates direct comparison between models and experimentally observed patterns of growth and gene expression.

In 3D, a third direction must be defined (*Kennaway and Coen, 2019*; *Whitewoods et al., 2020*). In MorphoGraphX, this can be done by combining the directions defined by different distance measures. The 3D Cell Atlas add-on (*Montenegro-Johnson et al., 2015*) combines several distance measures for radially symmetric structures such as root and hypocotyls. A Bezier curve is placed along the center in the longitudinal direction and combined with a surface mesh to obtain radial directions (*Figure 4B*). This also puts bounds on the radial direction (and also implicitly on the circumferential direction), which allows relative coordinates to be assigned to cells in addition to absolute values. The relative radial coordinate will follow the layer as the root narrows towards the tip. The relative distance between the central axis and the organ surface can then be used to annotate and classify 3D segmented cells in organs with a layered cellular organization. *Figure 4D* shows classification of layers with relative coordinates on an *Arabidopsis* root (*Montenegro-Johnson et al., 2015*). *Figure 4C* shows layer classification using absolute distance from a surface mesh (*Montenegro-Johnson et al., 2019*), which can be used as a starting point for layer classification in any organ.

The mature ovule in *Arabidopsis* shows a more complicated structure than a root or sepal, with five layers of integument cells encapsulating the nucellus that contains the embryo sac (*Schneitz et al., 1995*; *Vijayan et al., 2021*; *Figure 1—figure supplement 1*, *Figure 5*). In this example, combining different directions allows the establishment of organ-centric coordinates in the outermost layer of cells in the ovule (the outer integument). After segmentation and 3D mesh extraction in MorphoGraphX, directions normal to the surface (*Figure 5A*) were combined with those of a Bezier curve computed from a user-selected cell file (*Figure 5B and C*). Similar to the root data, relative coordinates facilitate the classification of cells into layers.

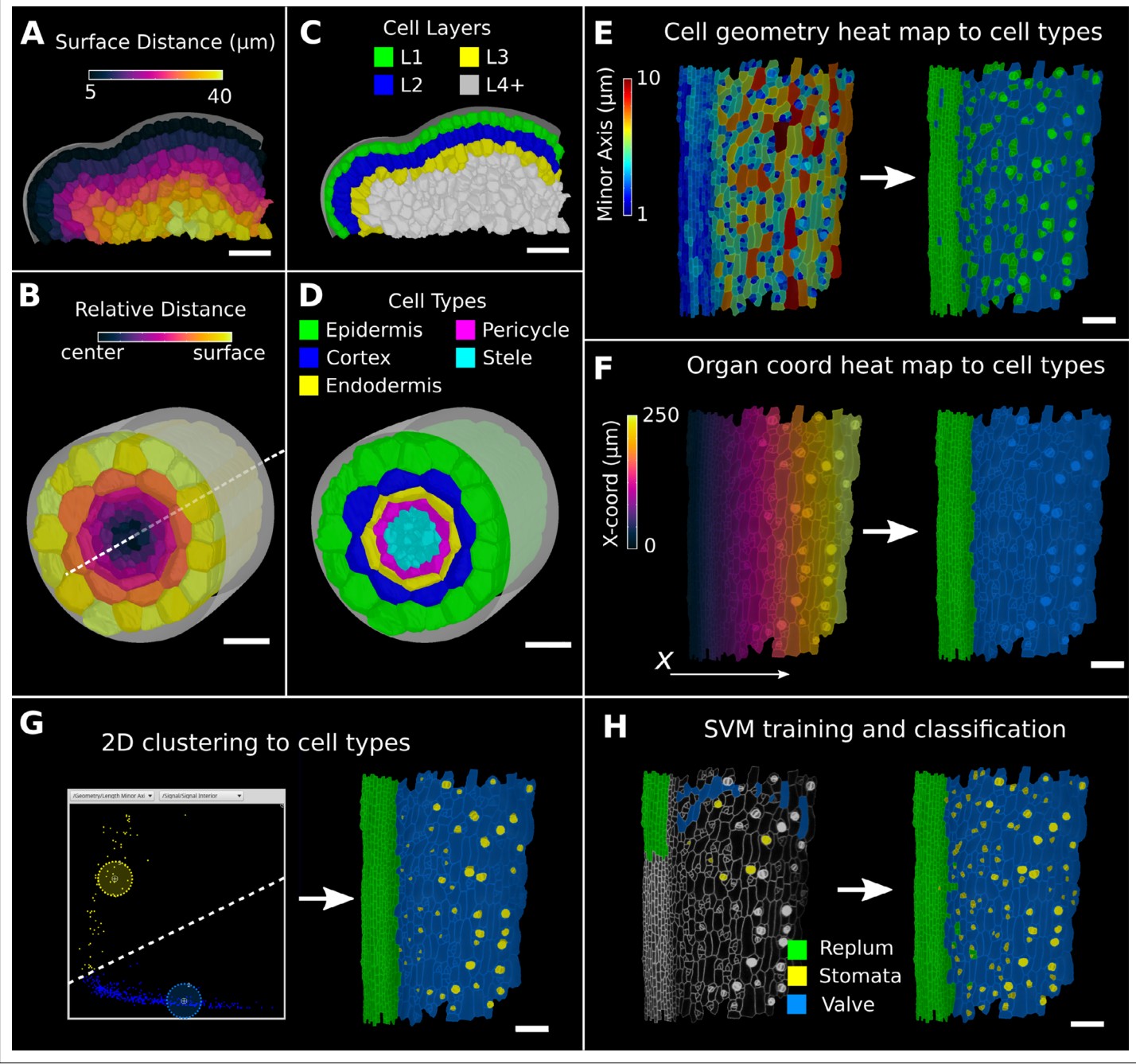

**Figure 4.** Methods to create organ coordinates for 3D meshes and label different cell types. (**A–D**) Organ coordinates and cell types for volumetric meshes. (**A**) Heat map of the surface distance for cell centroids in an *A. thaliana* shoot apical meristem. (**B**) For volumetric tissues, often a single direction is not enough to capture the geometry of the organ. Different methods can be combined such as a Bezier curve (white dashed line) with a surface mesh (gray) to create a heat map of the relative radial distance of cells in the *A. thaliana* root. (**C, D**) Organ coordinates can be used to assign cell-type labels as demonstrated in the 3D Cell Atlas plug-in for meristem and root. See also *Figure 4—figure supplement 1*. (**E–H**) Different methods to create cell-type labelings. (**E**) *A. thaliana* gynoecium (fruit epidermis) surface segmentation with a heat map of the length of the minor cell axis as obtained from a principal component analysis (PCA) on the cells' triangles. The heat values can be thresholded to assign two cell types. (**F**) The same principle can be used on organ coordinates that results in a clean separation of replum (green) and valve tissue (blue). (**G**) We generalized the 2D clustering approach of 3D Cell Atlas (see *Figure 4—figure supplement 1*) so that it can be used for any measure pair and on subset selections of cells. Shown is a 2D plot of the minor axis length (x-coord) and cell signal intensity (y-coord) on the valve tissue in (**F**). Manually assigning clusters can separate the stomata, which are typically smaller with higher signal values (yellow) and the remaining valve cells (blue) efficiently. See also *Figure 4—figure supplement 1* for 2D plots of all cells. (**H**) The support vector machine (SVM) classification is able to separate the three shown cell types in a higher dimensional space by

*Figure 4 continued on next page*

*Figure 4 continued*

using a variety of different measures and a relatively small training set. Scale bars: (**A–D**) 20 µm; (**E–H**) 50 µm. See also user guide Chapter 24 'Cell atlas and cell type classification' and tutorial videos S3 and S4 available at https://doi.org/10.5061/dryad.m905qfv1r.

The online version of this article includes the following figure supplement(s) for figure 4:

**Figure supplement 1.** Cell-type labeling methods and their use in the data analysis.

The Bezier curve defined the longitudinal direction, whereas the surface directions obtained from the organ surface mesh were used to compute perpendicular width and depth axes and distances. Cell volume and geometry acquired from 3D segmentation and mesh extraction (*Figure 1—figure supplement 1*) were calculated along the various directions of the organ axes and analyzed (*Figure 5D–G*).

Moving along the PD axis from 0 to 260 µm, we found variations in cell volume with a clear minimum at around 100 µm and a steady increase towards proximal and distal end (*Figure 5G*). Measuring the length, width, and depth of cells separately revealed the underlying cause for differences in cell volume between different PD regions of the outermost integument layer. At the proximal end (at 0 µm), the cell shape is relatively isotropic with similar values in cell length, width, and depth. Moving along the PD axis, cell anisotropy slowly increased, first mostly due the decreasing depth and width (until around 100 µm), later mainly due to the increased cell length (from about 150 µm). The increased cell length suggests a highly anisotropic growth along the longitudinal axis in this area. A potential proliferative region could be suspected in the region between 100 and 150 µm, where cell volume and length are the smallest. The quantification using the organ coordinates in this study allows spatial information to be linked to 3D cellular properties such as cell volumes and the associated shape anisotropies along different cell axes.

## Using positional information for automatic cell-type classification

Plant organs typically emerge as primordia consisting of undifferentiated tissue. Cells subsequently differentiate, acquiring a unique identity that depends on their location within the organ, via genetic processes that integrate spatial and environmental cues. Although cell differentiation is ultimately controlled by differential gene expression, it is often the case that cell fate can be predicted by geometry, even at very early stages (*Yoshida et al., 2014*). It is rare that cells with different cell types have identical morphological features.

MorphoGraphX supports a large variety of measures to quantify different features of cell morphology, including simple geometric measures (area, volume, perimeter, surface area, min and max axis), shape quantifiers (convexity, circularity, lobeyness, largest empty circle, aspect ratio), neighborhood measures (number of neighbors, variability), gene expression (average, total, near boundary), and cell network measures (betweenness centrality, betweenness current flow). Most measures can be used on time-lapse data to quantify changes over time (growth rates, gene expression changes, cell proliferation). For a complete list of the measures implemented in MorphoGraphX, see *Table 1* and *Table 2*. The modular architecture of the software also allows custom measures tailored to specific problems to be easily added through its plug-in interface. More sophisticated calculations, for example, the averaging of data over multiple samples, can be calculated externally in packages such as R and imported back into MorphoGraphX for visualization on segmented meshes. Here, the development of more complex data-flows is enabled by the use of a standardized attribute system to store and visualize cell data for both scalar values and tensor (directional) information.

During the segmentation process, MorphoGraphX assigns a unique label to each cell. When tracking cell lineages over multiple time points, a secondary label called the 'parent label' is provided. Other secondary labelings are also possible, for example, for cell type, cell layer, or zones within an organ. MorphoGraphX has several methods to assign these labels to classify cell types and layers. These labels can be assigned manually by interactively selecting cells or by employing a number of processes that use heat map measure data to assign secondary labels (*Figure 4E–H*). Positional information from the distance measures can be combined with measures of cell morphology and gene expression, where a secondary labeling can be used to provide additional context.

Since cells of a common biological cell type have similarities in one or more geometrical, positional, or gene expression attributes, the values of these attributes will often form a cluster, facilitating their automatic classification. An example can be seen in the 3D Cell Atlas add-on for MorphoGraphX

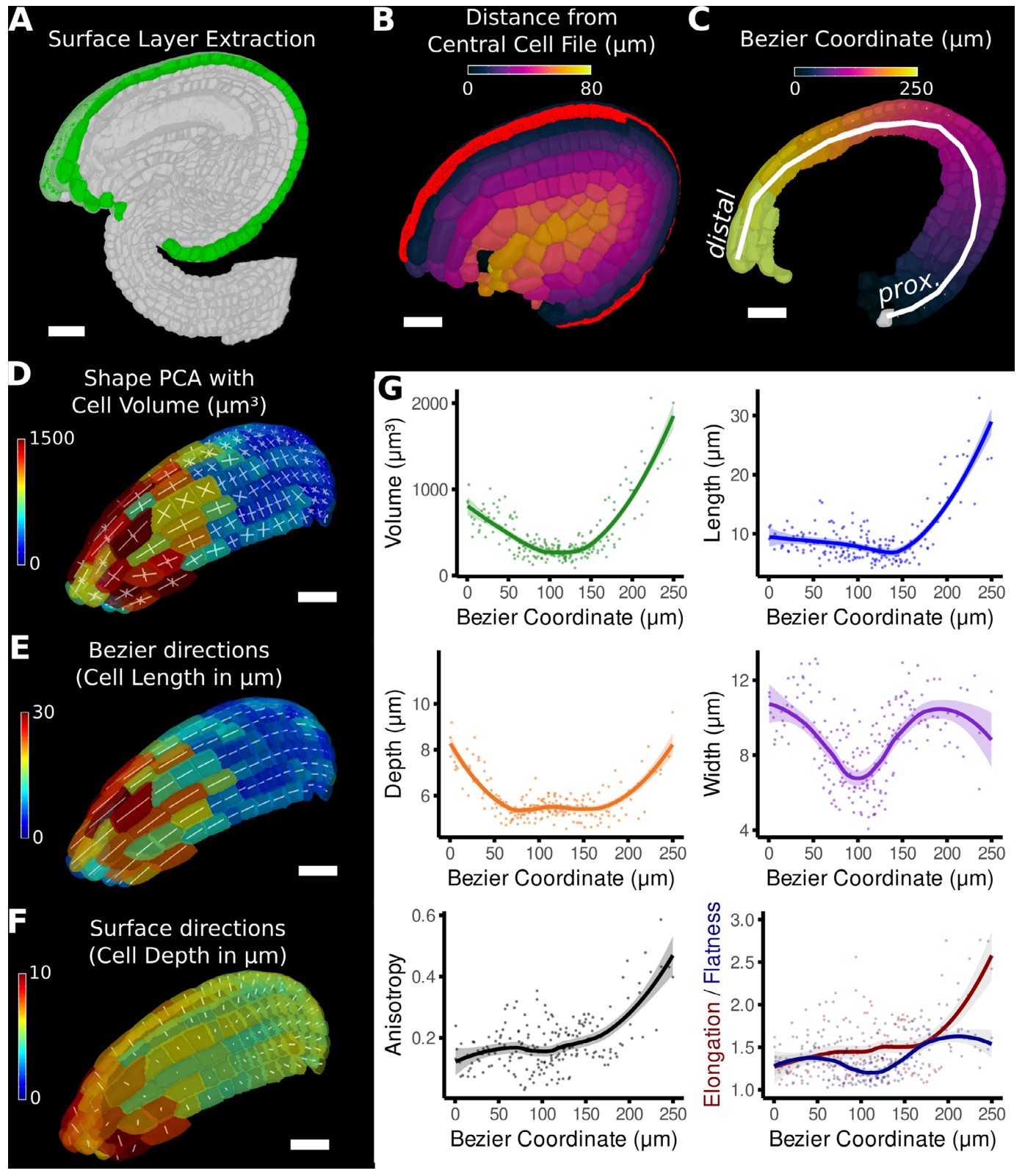

**Figure 5.** Quantification of volumetric cell sizes along organ axes in the outer layer of the outer integument of an *A. thaliana* ovule. (**A**) Extraction of cell layer of interest (colored in green) using an organ surface mesh. (**B**) Selection of the central cell file (in red) with cell distance heat map to exclude lateral cells (heat values >40 μm). (**C**) The centroids of the selected cells from (**B**) were used to specify a Bezier curve defining the highly curved organ axis from the proximal to the distal side. Heat coloring of the cells according to their coordinate along the Bezier. (**D–F**) Analysis of the cellular geometry

*Figure 5 continued on next page*

*Figure 5 continued*

in 3D. (**D**) Heat map of cell volume and the tensor of the three principal cell axes obtained from a principal component analysis on the segmented stack. (**E**) Bezier directions and associated cell length. (**F**) Directions perpendicular to the surface and associated cell depth. (**G**) Plots of the various cellular parameters relative to the Bezier coordinate. A few small cells at the distal tip of the integuments were removed from the analysis. Scale bars: 20 μm. See also user guide Chapters 21–24 and tutorial video S3 available at https://doi.org/10.5061/dryad.m905qfv1r.

(*Figure 4—figure supplement 1A–C*, *Montenegro-Johnson et al., 2015*) that clusters cells by relative radial distance and cell size to classify the cell layers of the root, hypocotyl, mature embryo, or other radially symmetric plant organs. To aid in optimizing cell clustering, MorphoGraphX offers a 2D interactive heat map, where information from two independent measures can be visualized, and clusters selected (*Figure 4G*, *Figure 4—figure supplement 1B*). These methods can be used repeatedly on subsets of cells to enable a classification of cells that differ across more than two features.

Multifeature classifications tasks can be solved automatically by machine learning approaches (*Cortes and Vapnik, 1995*) when provided with sufficient training data. Of particular relevance are support vector machines (SVMs) that have been used to classify cell types based on geometrical features of plant cells (*de Reuille and Ragni, 2017*; *Sankar et al., 2014*). MorphoGraphX provides a simple interface to the libSVM SVM library (*Chang and Lin, 2011*). Cells can be selected and classified into different cell types for use as training data (*Figure 4H*, *Video 2*). Any cell attribute or measure that can be quantified in MorphoGraphX can be used by the classifier. These include all the morphological and gene expression measures, time-lapse measures, positional information created via distance maps or other coordinate systems, and even custom measures created via plug-ins or calculated externally with R or MATLAB. Once trained on a small group of cells with the desired measures, the classifier can be used to classify all the cells in a sample (*Figure 4H*). After manual curation, the classification can then be used as additional training data, improving the power of the classifier. *Figure 4G and H* and *Figure 4—figure supplement 1F and G* show the cell types of the *Arabidopsis* gynoecium, which consists of the lateral valve tissues that are fused to the replum. Within the valve, stomata are homogeneously distributed, consistent with the uniform growth and differentiation of this tissue. In contrast, no stomata cells can be found within the replum, which is made up of smaller, more homogenously sized cells. Cell typing this organ can be useful to identify the region of the valve margin, where the fully matured fruit will dehisce to release the seeds (*Eldridge et al., 2016*; *Ripoll et al., 2019*).

## Mapping positional information through time

In the analysis of morphogenesis, many key quantifications such as growth depend on the ability to track samples through time. In MorphoGraphX, this can be done following cell segmentation by manually assigning parent labels to the second time point, a process that has been highly streamlined in the user interface for 2.5D surfaces. However for full 3D samples, or large 2.5D samples with multiple time points, this method can be cumbersome. One method to address this problem is to find a nonlinear coordinate transformation or deformation that maps all the points from one time point onto the next. Parent labeling can then be determined by mapping cell centers of the later time point to an earlier one, allowing the cell they came from to be identified. This can be used to directly assign lineage or to seed algorithms that use more involved methods, such as the minimization of the total distances between the mapped cells (*Fernandez et al., 2010*). In MorphoGraphX, to define a mapping between the meshes of two successive time points (*Figure 6A*), we have implemented a 3D deformation function based on scattered data point interpolation (using cubic radial-basis functions; *Duchon, 1977*; *Turk and O'Brien, 1999*). An initial transformation is computed based on a few preassigned landmarks by matching several cells with their parents in the previous time point (*Figure 6C*). A deformation field is then calculated that provides a mapping for all points in 3D. This is then used to assign parent labels in the second time point based on their closest match in the previous time point. Close to the landmark points, this mapping will be very accurate, with accuracy decreasing with distance from the landmarks. The decrease in accuracy away from landmarks is larger if the deformation between the time points is highly nonuniform. After assigning all the cells to their closest parent, the mapping is then verified by checking the correspondence of neighborhoods between each cell and its parent. The labeling for cells that do not match is cleared, and the process is repeated. This causes correctly labeled regions to 'grow' out from the initially placed landmarks until the entire

**Table 1.** Measures for cells segmented on surface projections (2.5D).

| Measure | Unit | Description |
|---|---|---|
| *Geometry* | | |
| Area | µm² | Area of the cell (sum of its triangle area) |
| Aspect ratio | - | Ratio of length major axis and length minor axis (see below) |
| Average radius | µm | Average distance from the center of gravity of a cell to its border |
| Junction distance | µm | Max or min distance between neighboring junctions of a cell |
| Length major axis | µm | Length of the major axis of the 2D shape analysis (computes a PCA on the triangle positions weighted by their area) |
| Length minor axis | µm | Length of the minor axis of the 2D shape analysis (computes a PCA on the triangle positions weighted by their area) |
| Maximum radius | µm | Maximum distance from the cell center to its border |
| Minimum radius | µm | Minimum distance from the cell center to its border |
| Perimeter | µm | Sum of the length of the border segments of a cell |
| *Lobeyness* | | |
| Circularity | - | Perimeter^2/(4*PI*Area) |
| Lobeyness | - | Ratio of cell perimeter to convex hull perimeter (1 for convex shapes) |
| Rectangularity | - | Ratio of cell area to the area of the smallest rectangle that can contain the cell (1 for rectangular shapes, lower values for irregular shapes) |
| Solidarity | - | Ratio of the convex hull area to the cell area (1 for convex shapes, higher values for complicated shapes) |
| Visibility pavement | - | 1-(visibility stomata) (see below) |
| Visibility stomata | - | Estimate of visibility in the cell 1 for convex shapes, decreases with the complexity of the contour |
| *Location* | | |
| Bezier coord | µm | Associated Bezier coordinate of a cell Requires a Bezier grid |
| Bezier line coord | µm | Associated Bezier coordinate of a cell Requires a Bezier curve |
| Cell coordinate | µm | Cartesian coordinate of a cell |
| Cell distance | µm/cells | Distance to the nearest selected cell (finds the shortest path to a selected cell through the cell connectivity graph, edge weights: Euclidean, cell number or 1/wall area) |

*Table 1 continued on next page*

*Table 1 continued*

| Measure | Unit | Description |
| --- | --- | --- |
| Distance to Bezier | µm | Euclidean distance to the Bezier curve or grid |
| Distance to mesh | µm | Euclidean distance to the nearest vertex in the other mesh |
| Major axis theta | ° | Angle between the long axis of the cell and a reference direction |
| Polar coord | °/µm | Polar coordinate around a specified Cartesian axis |
| *Network* | | |
| Neighbors | Count | Number of neighbors of a cell |
| Betweenness centrality | - | Computes the betweenness centrality of the cell connectivity graph Edges can be weighted by the length of the shared wall between neighboring cells |
| Betweenness current flow | - | Computes the betweenness current flow of the cell connectivity graph Edges can be weighted by the length of the shared wall between neighboring cells |
| *Signal* | | |
| Signal border | | Average amount of border signal in a cell |
| Signal interior | | Average amount of interior signal in a cell |
| Signal parameters | | Advanced and general process that allows the setting of parameters to compute different kinds of signal quantifications |
| Signal total | | Average amount of total (=border + interior) signal in a cell |
| *Other measure processes* | | |
| Mesh/lineage tracking/heat map proliferation | Cells | Proliferation |
| Mesh/cell axis/custom/custom direction angle | ° | Angle between a cell axis and a custom axis |
| Mesh/division analysis/compute division plane angles | ° | Angle between division planes and/or cell axes |

PCA: principal component analysis.

sample is correctly labeled (*Figure 6C*, *Video 3*). At each step, only correctly labeled cells remain. Sometimes the iterative cell labeling process can get stuck in highly proliferative areas where cells have divided repeatedly between time points. In this case, a few additional landmarks can be manually added at trouble spots. One significant advantage of the method is that incorrect cells remain unlabeled, making manual curation straightforward. Once all of the parents are assigned and have passed the neighborhood correspondence check, one can be assured that both the lineage and the underlying segmentations are correct.

Deformation functions can also be used to create animations of organ development from 2.5D or 3D time-lapse data. This requires two or more time points of a segmented mesh with corresponding cell lineages. The cell centers and/or junctions are used as the landmarks defining the deformation function that maps one mesh onto the other. Interpolating mesh vertices between stages creates

**Table 2.** Measures for meshes with volumetric (3D) cells.

| Measure | Unit | Description |
| --- | --- | --- |
| *Cell atlas* | | |
| Cell length (circumferential, radial, longitudinal) | μm | Cell length as determined by 3D Cell Atlas root (shoot rays from the cell center to the side walls to measure cell size along organ-centric directions) |
| Coord (circumferential, radial, longitudinal) | | 3D organ coordinates as determined by 3D Cell Atlas root |
| *Geometry* | | |
| Cell length (custom, X, Y, Z) | μm | Cell length along the specified direction (cell size is measured as in 3D Cell Atlas root [see above]) |
| Cell wall area | μm$^2$ | Total area of the cell wall |
| Cell volume | μm$^3$ | Volume of the cell |
| Outside wall area | μm$^2$ | Cell wall area that is not shared with another neighboring cell |
| Outside wall area ratio | % | Proportion of cell wall area that is not shared with a neighbor cell |
| *Location* | | |
| Bezier coord | μm | Associated Bezier coordinate of a cell Requires a Bezier curve or grid |
| Cell coordinate | μm | Cartesian coordinate of a cell |
| Cell distance | μm/cells | Distance to the nearest selected cell (finds the shortest path to a selected cell through the cell connectivity graph, edge weights: Euclidean, cell number or 1/wall area) |
| Distance to Bezier | μm | Euclidean distance to the Bezier curve or grid |
| Mesh distance | μm | Euclidean distance to the nearest vertex in the other mesh |
| *Network* | | |
| Neighbors | Count | Number of neighbors of a cell |
| Betweenness centrality | - | Computes the betweenness centrality of the cell connectivity graph Edges can be weighted by the length of the shared wall between neighboring cells |
| Betweenness current flow | - | Computes the betweenness current flow of the cell connectivity graph Edges can be weighted by the length of the shared wall between neighboring cells |

a smooth animation with as many intermediary steps as desired (*Figure 6—figure supplement 1*, *Video 4*). MorphoGraphX has a user-friendly pipeline to record animations directly from the GUI with options to adjust the camera angle and visualize cell lineages, heat maps, and cell outlines during the animation. Temporal smoothing of morphing animations created from more than two time points is achieved using Catmull–Rom splines to interpolate the position of mesh vertices over time (*Catmull and Rom, 1974*). Heat and signal values in the mesh, such as cell area, growth rates, or gene expression, can also be interpolated along with vertex positions.

In large cells, growth can vary significantly within the same cell (*Armour et al., 2015*; *Elsner et al., 2018*). As the deformation function provides a smooth mapping between time points, its gradient

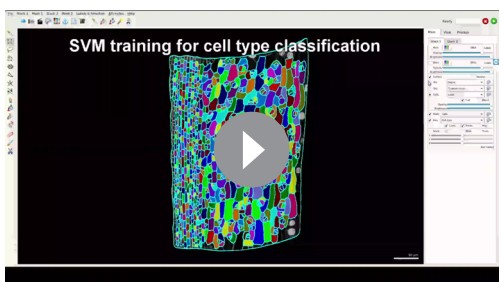

**Video 2.** Cell-type classification using support vector machines (SVMs). A selection of measures is calculated on a segmented mesh from an *Arabidopsis* gynoecium. A sampling of cells are then are labeled by hand and used as training data. The cell type for the rest of the cells is then predicted automatically.

https://elifesciences.org/articles/72601/figures#video2

can be used to create a continuous growth map at any point on a mesh. This enables the approximation of areal expansion and PDGs at a subcellular level, where the quality of the approximation is limited by the number and placement of landmarks (junctions). It is also possible to apply the process to subcellular landmarks, such as those obtained by tracking microbeads, as done previously for 2D images (*Armour et al., 2015*; *Elsner et al., 2018*). Our 3D implementation of this method has been used to compute growth directions on curved surface meshes (*Figure 6E*) and volumetric meshes (*Figure 7I*).

A comparison of the areal growth and PDGs calculated with deformation functions vs. the cell-based method is shown in *Figure 6D and E*. The deformation function captures differences in growth within single cells, as is often apparent in larger cells that straddle areas of fast and slow growth. *Figure 7A–I* shows a 3D time lapse of the *Arabidopsis* root where the deformation functions have been used to perform lineage tracking in 3D. It can be seen in *Figure 7D* that the four tissue types – epidermis, cortex, endodermis, and pericyle – all show a similar pattern of cell volume increase, with slow growth in the meristem and faster growth near the transition zone, which can also be seen when displayed as a function of the distance from the root tip (*Figure 7E*). Most of the volume growth occurs along the longitudinal direction and is reflected by the increase in cell length, which is highly synchronized due to the physical connections of the cell layers (*Figure 7F*). Growth along width and depth directions is less synchronized, but much smaller than the length increase, reflecting the strong anisotropic growth of this system (*Fridman et al., 2021*).

## Advanced geometric analysis

In addition to tools for creating organ coordinates and deformation functions, we have implemented a range of additional new processes in MorphoGraphX 2.0 for advanced image analysis.

### 3D lineage tracking and growth analysis

While MorphoGraphX was created to work with 2.5D surface projections, it now supports a complete set of tools for full 3D image processing, and in many samples advantages can be gained from combining both techniques. One example is automated cell lineage tracking, originally implemented for surfaces, which has now been extended to facilitate growth analysis in full 3D. Lineage tracking in 3D is a much harder task than on 2.5D surface images as 3D cellular meshes lack the cellular junctions that serve as material points for surfaces. However, in many cases, entire organs are well defined by their surfaces meshes, allowing landmarks on the surface to be used to construct a 3D deformation function to aid lineage tracking in full 3D. Surface landmarks can also be combined with 3D cell centers and/or face centers as material points to improve the internal resolution of the deformation functions for full 3D samples. These techniques allow the methods used to calculate growth rates and PDGs in 2.5D to be extended to full 3D (*Figure 7*; *Fridman et al., 2021*). Cell proliferation and most of the other measures can also be quantified from 3D time-lapse data (*Figure 7—figure supplement 1*). In addition to the automated tools, improved manual 3D parent labeling and the ability to relabel cells so that adjacent cells are always a different color aid in the manual curation of 3D lineage maps.

### Cell division analysis

One of the more advanced quantifications from time-lapse data is the analysis of cell divisions. As plant cells cannot move, cell division and growth are the main determinants of morphogenesis. MorphoGraphX has processes to identify dividing cells from time-lapse data and quantify the orientation of the division wall in both 2.5D and 3D (*Figure 8A–I*, *Figure 8—figure supplement 1*). In 2.5D, the best-fit line to the division wall is calculated (*Figure 8A and D*), whereas in 3D the best-fit plane

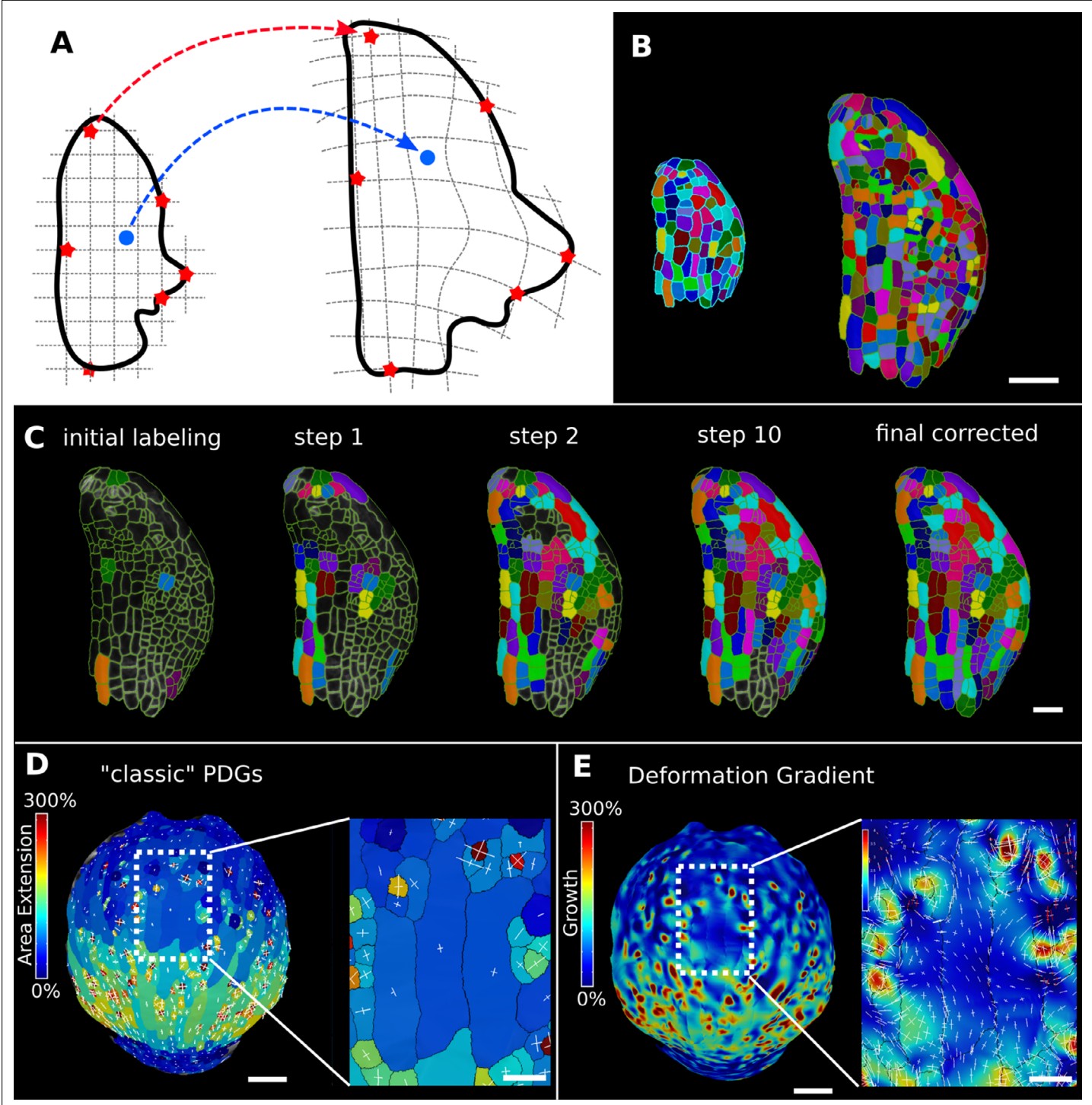

**Figure 6.** Deformation functions in MorphoGraphX. (**A**) Deformation functions allow a direct mapping of arbitrary points (blue) between two meshes. They require the definition of common landmarks (red stars). (**B, C**) Semi-automatic parent labeling using deformation functions. (**B**) Two consecutive time points of an *A. thaliana* leaf primordium segmented into cells. (**C**) The automatic parent labeling function requires the definition of a few manually labeled cells as initial landmarks. From this sparse correspondence, a mapping between the meshes can be created and new cell associations between the two meshes are added and checked for plausibility. With more cells found, the mapping between the meshes is improved for the next iteration. (**D, E**) Comparison of the classic principal directions of growth (PDGs) in (**D**) with the gradient of a deformation function computed using the cell junctions from a complete cell lineage in (**E**) on an *A. thaliana* sepal. The classic PDGs compute a deformation for each cell individually and are shown with a heat map of areal extension for each cell. In contrast, the deformation function is a continuous function on the entire mesh. Here, heat values are derived by multiplying the amount of max and min growth. Using the deformation function gradient subcellular growth patterns that were previously hidden are

*Figure 6 continued on next page*

*Figure 6 continued*

revealed, such as differential growth within a single giant cell. Scale bars: (**B, D, E**) 50 μm; (**C** and zoomed regions in **D** and **E**) 20 μm. See also user guide Chapter 17 'Mesh deformation and growth animation' and tutorial videos S1 and S2 available at https://doi.org/10.5061/dryad.m905qfv1r.

The online version of this article includes the following figure supplement(s) for figure 6:

**Figure supplement 1.** Deformation functions allow the interpolation of intermediate steps that can be turned into a continuous sequence or animation.

is used (*Figure 8B, C and G*). There are also measures to determine the asymmetry of the daughter cells. The use of positional information to give organ context is especially important for directional information, such as the orientation of cell division. Quantifying the orientation of the division plane is of little use without knowing how it relates to the developmental axes. Orientations can be computed with respect to the axes of a local coordinate system defined for the organ, along with its associated positional information (*Figure 8E and I*, *Figure 8—figure supplement 1A*). It is also possible to quantify how close cell divisions are to common division rules proposed in the literature, such as the shortest wall through the center of the cell including local minima (*Figure 8H*; *Besson and Dumais, 2011*; *Yoshida et al., 2014*; *Vaddepalli et al., 2021*) along the PDG (*Hejnowicz, 2014*), or rules based on network measures (*Jackson et al., 2019*).

## Cell connectivity networks

The organization of cells in organs may be analyzed through the extraction of cell connectivity networks from 2.5D or 3D segmented data. The physical associations between cells (cell-cell wall areas) can be extracted and converted into networks where they are analyzed using network measures (*Figure 8J–K*). Local measures such as the number of immediate neighbors (degree) can be calculated, along with more global measures, such as betweenness centrality based on the number of shortest paths cells lie upon, or random walk centrality (*Figure 8K*). These global measures are central to understanding how information flows within tissues (*Jackson et al., 2017*; *Jackson et al., 2019*). The use of these measures uncovered the presence of a global property in cellular organization within the *Arabidopsis* shoot apical meristem (SAM) (*Jackson et al., 2019*). Namely, the length of paths between cells is maximized, whereby cells that lie upon more shortest paths have a great propensity to divide, and the orientation of this division tends to leave the two daughter cells on the fewest number of shortest paths. Using this approach, the local geometric properties of cells can be related to the emergent global organization of cellular arrangements. Perturbation of cell shape in the *katanin1* mutant led to alterations in path length in the SAM, which correlated with defects in phyllotaxis (*Jackson et al., 2019*).

## Cell polarity

Cellular signals, such as proteins tagged with fluorescent reporters, can also be quantified in Morpho-GraphX. After segmenting a surface into cells by using a cell wall stain or marker line, a signal collected in a second channel can be projected onto the surface mesh, and the abundance, orientation,

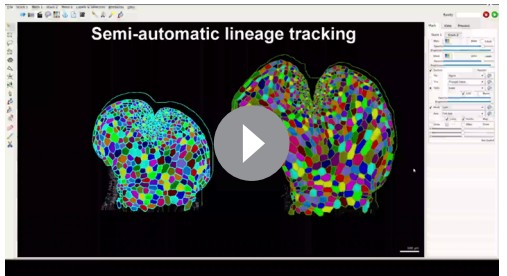

**Video 3.** Semi-automatic cell lineage tracking. A few cells from two time points from a time lapse of Marchantia are matched by hand. The other cells can then be determined automatically. Trouble spots can be overcome by selecting a few more cells by hand and continuing the process.

https://elifesciences.org/articles/72601/figures#video3

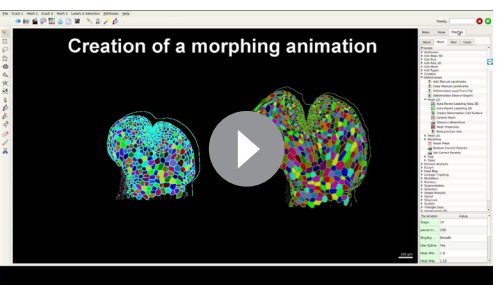

**Video 4.** Animating time-lapse data with deformation functions. A time lapse of Marchantia is used to demonstrate how deformation functions can be used to animate growth. Cells are colored by areal growth rate.

https://elifesciences.org/articles/72601/figures#video4

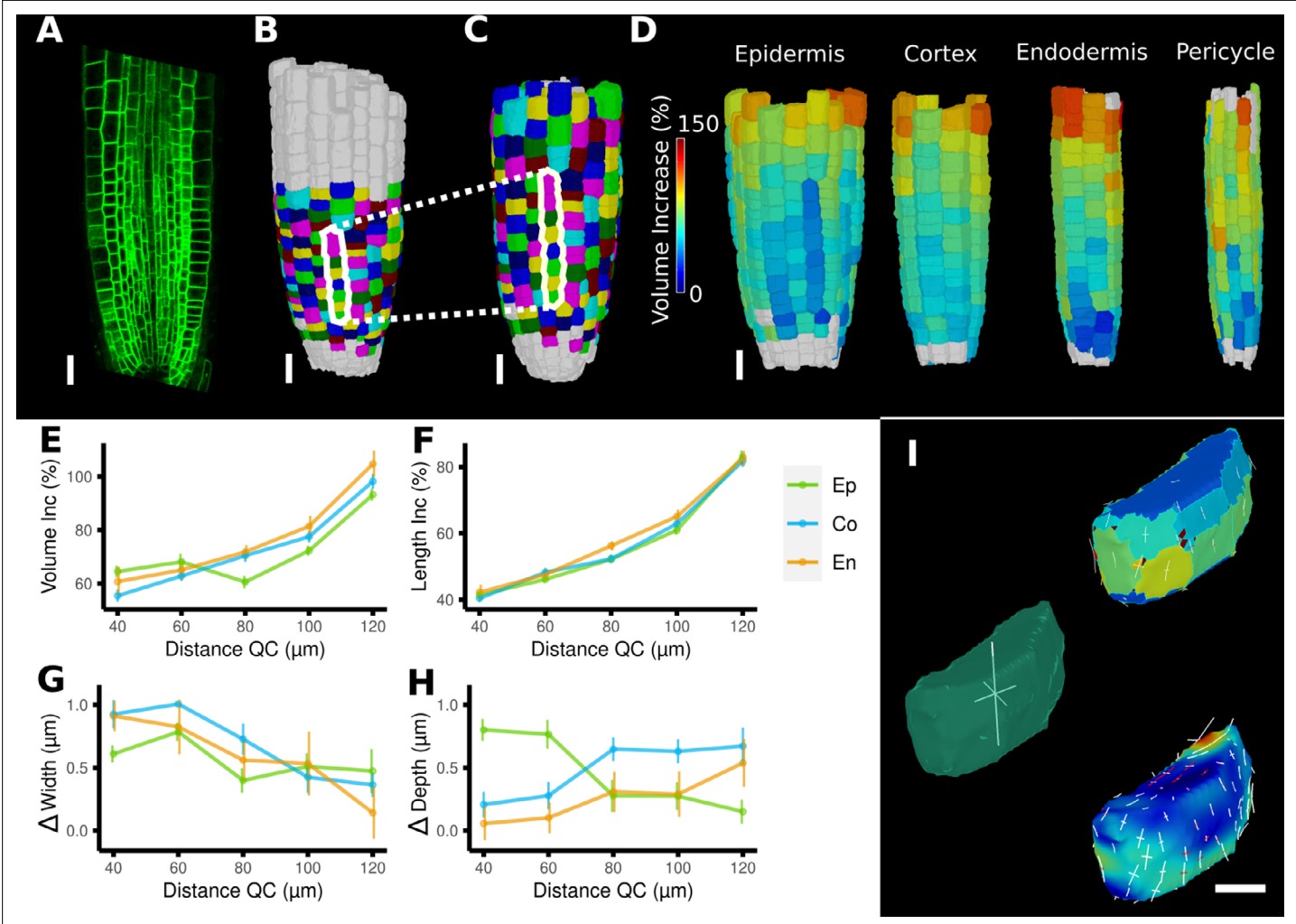

**Figure 7.** Time-lapse analysis and visualization of 3D meshes. (**A**) Cross section of the confocal stack of the first time point of a live imaged *A. thaliana* root. (**B, C**) The 3D segmentations of two time points imaged 6 hr apart. Shown are the cell lineages that were generated using the semi-automatic procedure following a manual correction. (**D**) Exploded view of the second time point with cells separated by cell types (see also *Figure 4D*). Cells are heat colored by their volume increase between the two time points. (**E–H**) Quantification of cellular growth along different directions within the organ. (**E**) Plot of the heat map data of (**D**). The cellular data was binned based on the distance of cells from the quiescent center (QC). Shown are mean values and standard deviations per bin. (**F–H**) Similarly binned data plots of the change in cell length (**F**), width (**G**), and depth (**H**). It can be seen that the majority of growth results from an increase in cell length. See *Figure 7—figure supplement 1* for a detailed analysis of the cells in the endodermis. (**I**) Different ways to visualize 3D growth demonstrated using a single cortex cell: principal directions of growth (PDGs) averaged over the entire cell volume (left), PDGs averaged over the cell walls projected onto the walls (top right), and subcellular vertex-level PDGs projected onto the cell walls (bottom right). Scale bars: (**A–D**) 20 μm; (**I**) 5 μm. See also user guide Chapter 21 'Mesh 3D analysis and quantification' and tutorial videos S6 and S7 available at https://doi.org/10.5061/dryad.m905qfv1r.

The online version of this article includes the following figure supplement(s) for figure 7:

**Figure supplement 1.** Time-lapse analysis of cellular geometry in the *A. thaliana* root endodermis.

and polarity of signals can be computed. Examples are the PIN-FORMED (PIN) auxin transporter report line (*Benková et al., 2003*) or the GFP:MBD (*Van Bruaene et al., 2004*) line that tags microtubules (MTs) (*Figure 8L–N*, *Figure 8—figure supplement 2*). For the quantification of cell polarity, MorphoGraphX implements a process where the projected signal along the cell border is binned based on its position in relation to the cell center to obtain its predominant direction and its intensity. *Figure 8L* shows an example of the PIN1 polarity quantification at the cell wall of surface segmented cells in the SAM. A similar quantification can be performed for 3D meshes as shown in *Figure 8N* and *Figure 8—figure supplement 2A and B*, where we computed the PIN2 polarity in epidermis and cortex cells of an *Arabidopsis* root. Again, this directional information can be combined with the

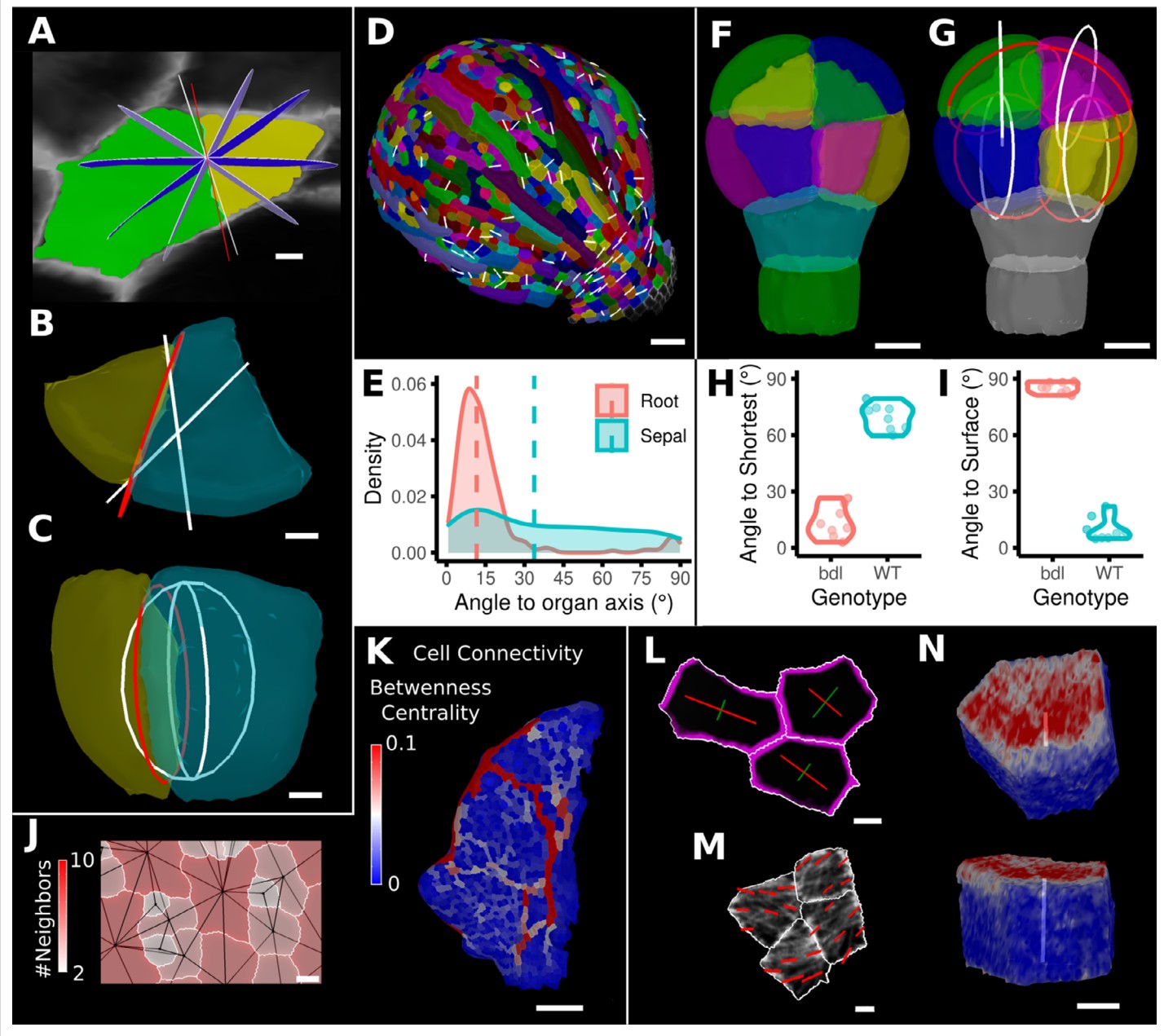

**Figure 8.** Advanced data analysis and visualization tools. (**A**) Division analysis of a cell from a surface segmentation of an *A. thaliana* sepal. A planar approximation of the actual plane is shown in red and other potential division planes in white/blue. The actual wall is very close to the globally shortest plane. (**B, C**) Top and side views of a recently divided 3D segmented cell. The daughter cells are colored yellow and cyan. The red circle depicts the flat approximation plane of the actual division wall. The two white rings depict the two smallest area division planes found by simulating divisions through the cell centroid of the mother cell (i.e., the combined daughter cells). (**D**) Visualization of the actual planes (white lines) between cells that divided into two daughter cells in the *A. thaliana* sepal. (**E**) Density distribution and median (dashed line) of the angle between the division plane and the primary organ axis in sepal (see **D**) and root (see *Figure 8—figure supplement 1A*). The division in sepals is less aligned with the organ axis. (**F**) Half of an *A. thaliana* wildtype embryo in the 16-cell stage. This view shows that the divisions leading to this stage are precisely regulated to form two distinct layers in the embryo. (**G**) A visualization of the actual planes (red circles) and the shortest planes (white circles) in the wild type. Cells are colored according to the label of the mother cells. (**H, I**) Violin plots of quantifications of the planes show that the wild type does not follow the shortest wall rule unlike the auxin-insensitive-inducible bdl line RPS5A>>bdl. The bdl divisions are almost orthogonal to the organ surface (see *Figure 8—figure supplement 1B, D, E*), whereas the wild type divides parallel to the surface. Consequently, the bdl fails to form a distinct inner layer. (**J, K**) Cellular connectivity network analysis. (**J**) Cell connectivity network analysis on a young *A. thaliana* leaf. Cells are heat colored based on the number of neighbors, edges in the cell connectivity graph are shown in black. (**K**) Heat map of betweenness centrality. The betweenness reveals pathways that might be of importance for information flow, potentially via the transport of auxin. (**L–N**) Cell-based signal analysis. (**L**) Analysis of cell polarization on a surface mesh.

*Figure 8 continued on next page*

*Figure 8 continued*

(**M**) Microtubule signal analysis on a surface mesh. (**N**) Top and side views of a cell polarization analysis on a volumetric mesh (root epidermis PIN2, see *Figure 8—figure supplement 2A–D* for details). Scale bars: (**A, B, C, L, M**) 2 μm; (**D**) 50 μm; (**F, G, J, N**) 5 μm; (**K**) 100 μm. See also user guide Chapter 25 'Cell division analysis.', Chapter 18 'Quantifying signal orientation', and Chapter 21.7 'Signal orientation for 3D meshes'.

The online version of this article includes the following figure supplement(s) for figure 8:

**Figure supplement 1.** Details of the cell division analysis examples from *Figure 8*.

**Figure supplement 2.** Example analyses of cell polarity and microtubule signals of the data shown in *Figure 8M and N*.

organ coordinates to compute the angle between cell polarity and the organ axis (*Figure 8—figure supplement 2C*). Another example is the quantification of MT alignment using our implementation of FibrilTool (*Boudaoud et al., 2014*) that has been adapted for processing on surfaces. After projecting the MT signal onto the surface, the alignment direction and strength of the signal can be quantified at the subcellular level (*Figure 8M*) or for entire cells (*Figure 8—figure supplement 2E*). Again, this information can be interpreted using organ coordinates as we demonstrate on cells of a SAM that tend to have their MTs aligned circumferentially from the meristem center (*Figure 8—figure supplement 2E,F*).

## 3D visualization and interactive tools

MorphoGraphX has a flexible rendering engine that can handle meshes containing millions of triangles. It supports the independent rotation and translation of different stacks and meshes in the same world space and the ability to render both voxel and geometric data together with blending and transparency. It has adjustable clipping plane pairs and a bendable Bezier cutting surface that can be used to look inside 3D samples, and an interface to support the creation of animations (*Video 5*). However, visualizing and interacting with 3D data on a 2D computer screen still remains a challenge. A particular problem is the validation and correction of 3D segmentations of organs as internal cells are obscured by outer layers. 3D segmentation curation and correction is a bottleneck when developing training sets for deep learning tools, and MorphoGraphX has become a useful tool for this purpose (*Wolny et al., 2020*). In addition to clipping planes, exploded views are a commonly used method to visualize the internal structure of multicomponent 3D objects (*Li et al., 2008*), which can be used on mesh data in MorphoGraphX. These are particularly useful for visualizing the internal structure of entire organs with small cell numbers such as embryos (*Figure 8—figure supplement 1C*) and can be used in combination with multiple clipping planes for larger samples. To add biological meaning to the exploded visualization, cells can be bundled by their parent or cell-type label to visualize key aspects of biological data sets such as cell divisions or to separate organs into cell layers or by cell type (*Figure 7D*, *Figure 8—figure supplement 1C*). Furthermore, cells sharing the same cell-type label can be easily manually selected, moved, or deleted for improved visualization of specific tissues and groups of cells. These processes can also aid user interactions, making cell selection and annotation more straightforward when users are curating 3D cellular segmentations and lineage maps.

In addition to mesh editing tools, Morpho-GraphX has several tools to edit voxel data. The

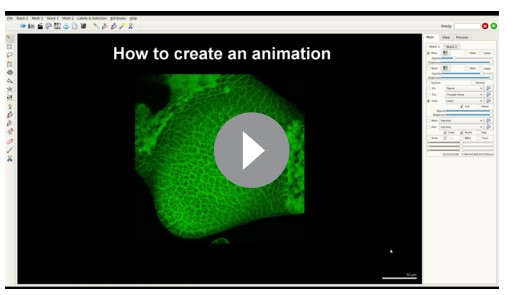

**Video 5.** Creating an animation. Key frames are saved and used to provide steps for an animation that can then be played back.

https://elifesciences.org/articles/72601/figures#video5

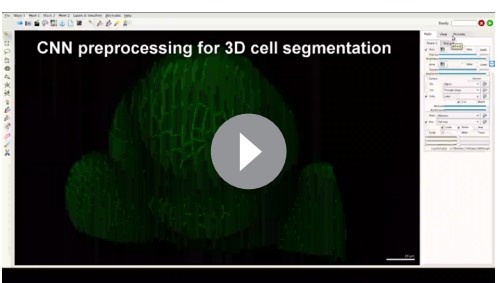

**Video 6.** Convolutional neural network (CNN) cell wall prediction and segmentation. Using an *Arabidopsis* flower meristem, a CNN is first used to improve the cell wall signal. The meristem is then segmented into cells, and a 3D mesh created.

https://elifesciences.org/articles/72601/figures#video6

simplest is an eraser tool that can be used to remove portions of the stack that would otherwise interfere with processing. An example is the digital deletion of the peripodial membrane overlying the *Drosophila* wing disc, which needs to be removed to allow for the extraction of the organ's surface (*Aegerter-Wilmsen et al., 2012*). A typical workflow for 3D segmentation starts with a 3D image of a cell boundary marker. This is then preprocessed with operations such as blurring to reduce noise or background removal filters, before segmentation with algorithms such as the Insight Toolkit's (ITK) Morphological Watershed filter (https://www.itk.org). More recently, deep learning methods with CNNs have been developed to predict cell boundaries, such as

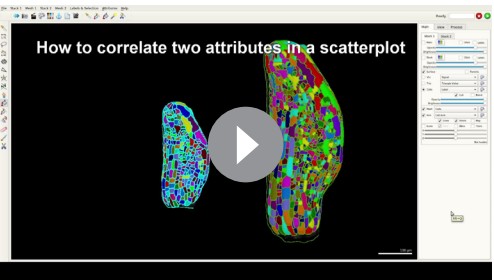

**Video 7.** Calling R from MorphoGraphX. A time lapse of an *Arabidopsis* leaf is used to demonstrate how to create heat map data for analysis and plotting in R. https://elifesciences.org/articles/72601/figures#video7

the 3D U-Net model (*Çiçek et al., 2016*; *Ourselin et al., 2016*), that can improve the stacks for downstream segmentation. The modular structure of MorphoGraphX has allowed us to interface an implementation of the 3D U-Net model developed by *Eschweiler et al., 2019*. This enables the interactive use of the CNN boundary prediction tool from within MorphoGraphX (*Video 6*) and simplifies experimentation with different networks or downstream segmentation strategies. Several networks are currently implemented (*Eschweiler et al., 2019*; *Eschweiler et al., 2021*; *Wolny et al., 2020*), although the add-on should work with any libtorch traced model. It also avoids the requirement to set up a full Python environment and comes stand-alone as a package built for the most common nVidia GPU architecture.

Once data is segmented, it often requires some manual correction before it is ready for final analysis in a chosen coordinate system. MorphoGraphX has interactive tools that operate on voxel data both to combine and split labels (cells), although typically it is easier to oversegment and combine, rather than undersegment and split (*Video 6*). This can be used to correct segmentations, which can then be used to help train deep learning networks to further improve automatic segmentation (*Vijayan et al., 2021*). In this context, MorphoGraphX has been used to segment and curate *Arabidopsis* ovule data to create ground truth for confocal prediction networks (*Wolny et al., 2020*).

## Software design

The internal architecture of MorphoGraphX has been designed to make it easily extendable, while retaining the speed of the fully compiled, statically typed language C++. The relatively small visualization and data management core is augmented with processes that are loaded dynamically at startup and provide almost all of the software's functionality. MorphoGraphX has grown to provide a wealth of custom processes for 2.5D and 3D image processing and coordinate system creation. Additionally, it has become a platform to integrate published tools and methods that have no visual interface to the data of their own, which increases their accessibility and ease of exploration for biologists. Examples include the previously mentioned deep learning tools and the processes that interface to the Insight Toolkit (ITK) C++ image processing libraries. This greatly reduces the learning curve required to use ITK tools, such as the popular Morphological Watershed filter, which can be run from MorphoGraphX with a click of the mouse. For more advanced ITK pipelines, we have recently integrated an open-source tool that provides an XML Pipeline Wrapper for the Insight Toolkit (XPIWIT) that allows the development of ITK image processing pipelines interactively without any C++ programming required (*Bartschat et al., 2016*). These pipelines can be packaged into processes and called directly from the MorphoGraphX GUI, allowing the easy exploration of complex ITK image processing pipelines. An example is the 'Threshold of weighted intensity and seed-normal gradient dot product image' (TWANG) pipeline (*Stegmaier et al., 2014*), a fast parallel algorithm for nuclear segmentation. A selection of pipelines along with the XPIWIT software is bundled as an add-on for MorphoGraphX and requires no additional installation.

Another example of software integration with popular open-source tools is the processes we have developed to interface with R (*Video 7*) that provide plots for basic statistical analysis on attributes

created in MorphoGraphX, including positional information provided by organ-centric coordinate systems. This simplifies the creation of the most commonly used plots without the need for export files and the mastery of ggplot. In addition to directly linking to C++ libraries, MorphoGraphX can be scripted with Python, allowing repetitive functions to be performed in batch, and the possibility to integrate Python-based tools. Operations interactively performed in MorphoGraphX are written to a Python log file for reproducibility and logging, and to allow easy cut-paste script creation.

Since its inception, a major focus of MorphoGraphX has been in the creation, manipulation, and visualization of geometry in the form of surface meshes for 2.5D and full 3D cellular meshes. This is in addition to the 3D voxel source data from microscopy images. There is currently very little software available that can handle both and enable the interaction between the two. Most image processing software work only with voxels, whereas most computer graphics, animation, and engineering software deal only with meshes. If the goal is simply to segment 2D or 3D cells, there are many options available, for example, Fiji (*Schindelin et al., 2012*) or ilastik (*Berg et al., 2019*). If the processing of surface projections is required, advanced geometrical quantification like cell division analysis, or annotation with custom designed coordinate systems, then MorphoGraphX should be considered.

## Conclusion

Similar to sequencing data, geometric data on the shape and sizes of hundreds or thousands of cells is of limited value without annotation. For many developmental questions, the spatial context for information on cell shape, division, and gene expression is paramount. However, it is not enough to know the 3D position of cells, but rather their position in a coordinate system relative to the developing tissue or organism. These coordinates typically reflect the developmental axes of the organism or tissue, allowing the direct comparison of cell and organ shape changes with the gene expression controlling their morphology. In addition to putting data in a mechanistic context, organ-centric relative coordinates can be used to compare samples with different morphologies (*Thompson, 1942*), such as different mutants, or even in different species (*Kierzkowski et al., 2019*). This also applies to changes in morphology over time, where organ coordinate systems can be used to determine the correspondence between cells at different stages of development. Several tools have been published that successfully harness organ-centric coordinates for specific problems, for example, in roots and hypocotyls (*Montenegro-Johnson et al., 2015*; *Schmidt et al., 2014*), the shoot apex (*Montenegro-Johnson et al., 2019*), and the *Arabidopsis* ovule and similar shaped organs (*Vijayan et al., 2022*). MorphoGraphX provides a generalized framework to create such tools by enabling the development of coordinate systems customized to the particular organ or organism of interest. Possibilities vary from simple distance-based systems to analyze leaf growth on surfaces from time-lapse images (*Kierzkowski et al., 2019*) to more involved methods for more complex organs in full 3D from fixed samples (*Vijayan et al., 2021*).

MorphoGraphX is unique in that it is the only software that we are aware of that allows image processing on surface meshes, which we informally refer to as 2.5D image processing. These meshes are most often created from 3D confocal images of a cell boundary signal. Images obtained from the microscope in proprietary formats are typically loaded into Fiji or ImageJ with BioFormats/LOCI-Tools (*Linkert et al., 2010*), and then converted to TIFF for import into MorphoGraphX. Denoising of images and other preprocessing can be performed before import; however, most common preprocessing steps are now available directly in MorphoGraphX. Although originally targeted at 2.5D image processing, the necessity to handle full 3D image processing has meant that it has become convenient to implement a wide array of processing filters for full 3D in MorphoGraphX, many of which are GPU accelerated. The number of processes available in MorphoGraphX has swelled to over 540 in the current version compared to just over 160 in the previously published version (*Barbier de Reuille et al., 2015*). This includes a comprehensive toolkit for cell shape analysis, growth tracking, cell division analysis, and the quantification of polarity markers, both on 2.5D image meshes, and for full 3D. All of these measures can be calculated and stored within the mesh, or exported to files for further processing with other software. Custom measures can also be calculated and imported for visualization within MorphoGraphX. Cell shape measures in combination with positional information provide a powerful framework for cell-type classification, both with machine learning methods (*Figure 4G and H*) and clustering techniques (*Montenegro-Johnson et al., 2019*). Although organ-centric positional information provides important features for cell classification, it is unavailable in

most machine learning cell classification software that typically can only deal with voxel information and limited annotation.

A key strength of MorphoGraphX is that it offers both manual and automatic tools for segmentation, lineage tracking, and data analysis. Although fully automated methods are improving, streamlined methods for manual and semi-automatic segmentation and analysis provide a path to completion for many samples where the automatic methods are 'almost' good enough. For example, the automatic lineage tracking now available in MorphoGraphX benefits from the streamlined tools we developed previously to do the process manually as these are now used to correct and fill in missing portions when the automatic segmentation is incomplete. This reflects the interactive nature of MorphoGraphX and its focus on low-throughput but high-quality datasets.

As more and more imaging datasets are becoming available for community use, their annotation with positional information and gene expression data will be critical to help understand how the cell-level action of different genes and genetic networks is translated into the 3D forms of tissues and organs of different species (https://www.plantcellatlas.org). In this context, MorphoGraphX provides a tool set to help maximize the attainable information from these datasets in an accessible platform tailored to the experimental biologist.

## Materials and methods

### Software availability

MorphoGraphX is open-source software and runs on Linux and Windows. Binaries and source code can be downloaded from: https://www.MorphoGraphX.org/software.

There are binaries available for recent versions of Ubuntu, as well as Windows. We recommend Linux as some add-ons are not available for Windows. For Linux, we provide a Cuda version for machines with a compatible nVidia graphics card and a non-Cuda version for those without. Currently there is only a non-Cuda version for Windows. Although there is no Mac version, some have had success running it in a virtual machine.

Support for the software can be found on the Help page of the MorphoGraphX website and the MorphoGraphX user forum on forum.image.sc.

With the growing number of processes in MorphoGraphX, the complexity of the software has increased. We aim to provide novel users with sufficient documentation to do their first steps in MorphoGraphX. We previously published a detailed guide on how to generate cellular segmentations and do basic quantifications as shown in *Figure 1* (*Strauss et al., 2019*). MorphoGraphX 2.0 also comes with an extended manual that contains step-by-step guides for all workflows presented in this article (*Table 3*).

### Data acquisition

#### *Arabidopsis* flower meristem (*Figure 1*)

pUBQ10::acyl:TDT (*Segonzac et al., 2012*) and DR5v2::n3eGFP (*Liao et al., 2015*) were crossed. F3 double homozygote line was used for imaging. Floral meristems were dissected from 2-week-old plants grown on soil under the long-day condition (16 hr light/8 hr dark), at 20°C ± 2°C using injection needle. Dissected samples were cultivated in 1/2 Murashige and Skoog medium with 1% sucrose supplemented with 0.1% plant protective medium under the long-day condition (16 hr light/8 hr dark), at 20°C ± 2°C. Confocal imaging was performed with Zeiss LSM800 with a 40× long-distance water immersion objective (1 NA, Apochromat). Excitation was performed using a diode laser with 488 nm for GFP and 561 nm for TDT. Signal was collected at 500–550 and 600–660 nm, respectively. Images of three replicates were obtained every 24 hr for 4 days.

#### *Arabidopsis* ovule (*Figure 5A–F*, *Figure 1—figure supplement 1*)

Data previously published in *Vijayan et al., 2021*.

**Table 3.** MorphoGraphX workflows.

| Workflow | Description | Used in figure | User guide chapter |
|---|---|---|---|
| *MorphoGraphX legacy workflows* | Processes and pipelines introduced in MorphoGraphX 1.0 (*Barbier de Reuille et al., 2015*) | | |
| Surface segmentation | Creating a surface mesh, projecting epidermal signal and segmentation | *Figure 1A–C* | The chapters at the beginning of the user guide deal with these basic topics: from chapter 1 'Introduction' to 12 'Attribute maps & data export' |
| 3D segmentation | Creating volumetric segmentation using ITK watershed | *Figure 1—figure supplement 1A and B* | 20 '3D segmentation' |
| Parent labeling | Cell lineage tracing between two subsequent segmented time points of the same sample | *Figure 1D and E* | 13 'Lineage tracking' |
| Cell geometry heat maps | Creating heat map of cellular data | *Figure 1D, Figure 1—figure supplement 1C* | 10 'Cell geometry quantification' |
| Time-lapse heat maps | Creating heat maps using two parent labeled time points | *Figure 1E* | 14 'Comparing data from two time points' |
| Growth directions | Computing growth directions using time-lapse data | *Figure 1E* | 15 'Principal directions of growth (PDGs)' |
| *MorphoGraphX 2 workflows* | New processes and pipelines introduced in this article | | |
| Different methods of creating organ coordinates | Using world coordinates (X, Y, Z) | *Figure 2A* | 23.3 'Further types of organ coordinates' |
| | Using polar coordinates | *Figure 3G–I* | 23.3 'Further types of organ coordinates' |
| | Using Bezier line or grid | *Figure 2C* | 23.2 'Bezier line and grid' |
| | Using cell distance heat maps | *Figure 2D* | 23.1 'The cell distance measure' |
| | Using a separate organ surface mesh | *Figure 4A* | 23.3 'Further types of organ coordinates' |
| Different methods of creating cell-type labelings | Using heat maps or organ coordinates | *Figure 4E and F* | 24.4 'Cell type classification using a single heat map' |
| | Using 2D clustering | *Figure 4G* | 24.5 'Cell type classification using two measures' |
| | Using support vector machines (SVMs) | *Figure 4H* | 24.6 'Cell type classification using SVMs' |

*Table 3 continued on next page*

*Table 3 continued*

| Workflow | Description | Used in figure | User guide chapter |
| --- | --- | --- | --- |
| Organ directions | Deriving directions from coordinates or Bezier curves | *Figure 3B and H* | 16 'Custom axis directions' and 21.6 'Custom directions for 3D meshes' |
| Combining directions | Combining different kind of organ coordinates or directions | *Figure 4B*, *Figure 5D–F* | 24.1 'Cell Atlas root" and 21.6 'Custom directions for 3D meshes' |
| Semi-automatic parent labeling | For lineage tracing | *Figure 6A–C* | 17.2 'Semi-automatic parent labeling' |
| Morphing animations | | *Figure 6—figure supplement 1A and B* | 17.4 'Morphing animations' |
| 3D growth analysis | | *Figure 7D–I* | 21.3 'Change maps 3D' and 21.4 'PDGs 3D' |
| Division analysis | Analyzing cell divisions | *Figure 8A–I, Figure 8—figure supplement 1A and E* | 25 'Division analysis' |
| Cell connectivity analysis | Analyzing cellular connectivity networks | *Figure 8J and K* | - |
| 3D visualization | Exploded views | *Figure 8—figure supplement 1C* | 21.5 '3D visualization options' |
| Signal quantification | Quantifying signal amount and direction | *Figure 8L–N, Figure 8—figure supplement 2A–F* | 18 'Quantifying signal orientation' and 21.7 'Signal orientation for 3D meshes' |

*Arabidopsis* root (*Figure 2A*, *Figure 4B and D*, *Figure 4—figure supplement 1A and C*, *Figure 7A–D and I*, *Figure 7—figure supplement 1A–D*, *Figure 8—figure supplement 1A*)

Root imaging

pUBIQ10::H2B-RFP pUBQ10::YFP-PIP1;4 was described previously (*von Wangenheim et al., 2016*). The seeds were stratified for 1 day at 4°C, grown on 1/2 Murashige and Skoog medium with 1% sucrose under the long-day condition (16 hr light/8 hr dark) at 20°C ± 2°C. Confocal imaging was performed with Zeiss LSM780 with two-photon laser (excitation 960 nm) with a band pass filter 500/550 nm for YFP. Images of three replicates were obtained.

*Arabidopsis* mature embryo (*Figure 2C*)

*A. thaliana* Col-0 seeds were sterilized in 70% ethanol with Tween20 for 2 min, replaced with 95% ethanol for 1 min and left until dry. Seeds were placed on the Petri plates containing 1/2 MS medium including vitamins (at pH 5.6) with 1.5% agar and stratified at 4°C for 3 days in darkness. Next, seeds were imbibed for 3 hr, and the mature seed embryo was isolated from the seed coat. For live imaging, the embryos were stained with propidium iodide 0.1% (Sigma-Aldrich) for 3 minutes and imaged with Leica SP8 laser scanning confocal microscope with a water immersion objective (×20). Excitation wavelengths and emission windows were 535 nm and 617 nm. Confocal stacks were acquired at 1024 × 1024 resolution, with 0.5 µm distance in Z-dimension. Images were acquired at 48 hr intervals and samples were kept in a growth chamber under long-day condition (22°C, with 16 hr of light per day) between imaging.

### *Marchantia* time lapse (Video 3)

*Marchantia polymorpha* gemmaling Cam-1 PM::GFP reporter line (*Boehm et al., 2017*) were transferred on a Petri plate containing 1/2 Gamborg's B5 medium including vitamins (pH 5.5) with 1.2% agar and grown for 24 hr. For live imaging, the gemmaling were imaged with Leica SP5 laser scanning confocal microscope with a water immersion objective (×25/0.95). Excitation wavelengths and emission windows were 488 nm and 510 nm. Confocal stacks were acquired at 1024 × 1024 resolution, with 0.5 µm distance in Z-dimension. Images were acquired at 24 hr intervals, and samples were kept in a growth chamber under constant light between imaging. For the move, we selected a representative sample from six total replicates. To quantify the cell area, change, and anisotropy, the fluorescence signal was segmented and semi-automated parent labeling was performed to couple the cells at two successive time points. Heat maps are displayed on the later time point (after 24 hr of growth). Scale bars are displayed on the image.

*Arabidopsis* sepal (*Figure 2D and E*, *Figure 2—figure supplement 1A and B*, *Figure 6D and E*, *Figure 6—figure supplement 1B*, *Figure 8A and D*)

Data previously published in *Hervieux et al., 2016*.

*Arabidopsis* leaf (*Figure 3A–E*, *Figure 6—figure supplement 1A*, *Figure 8J and K*)

Data previously published in *Kierzkowski et al., 2019*.

Tomato shoot apical meristem (*Figure 3F–I*)

Data previously published in *Kierzkowski et al., 2012*.

*Arabidopsis* shoot apical meristem (*Figure 4A and C*, *Figure 4—figure supplement 1D*)

Data previously published in *Montenegro-Johnson et al., 2019*.

*Arabidopsis* gynoecium (*Figure 4E–H*) and leaf (*Figure 6B and C*)

pUBQ10::acyl:YFP has been described previously (*Willis et al., 2016*). Plants were cultivated on soil under the long-day condition (16 hr light/8 hr dark) and 20°C ± 2°C. Flowers at post-anthesis stage

from 5-week-old plants were dissected with fine tweezers to remove sepals and stamens to expose gynoecium and mounted on the 60 mm plastic dish filled with 1.5% agar. Confocal imaging was performed with a Zeiss LSM800 upright confocal microscope, equipped with a long working-distance water immersion objective ×40 (1 NA, Apochromat). Excitation was performed using a diode laser with 488 nm for YFP, and the signal was collected between 500 and 600 nm. For both organs, images of three replicates each were obtained.

### *Arabidopsis* embryo (*Figure 8B,C,F and G*, *Figure 8—figure supplement 1B–E*)

Data previously published in *Yoshida et al., 2014*.

### *Arabidopsis* shoot apical meristems for PIN1 and MT (*Figure 8L and M*, *Figure 8—figure supplement 2E*)

pUBQ10::acyl:TDT (*Segonzac et al., 2012*) and GFP:MBD (*Van Bruaene et al., 2004*) were crossed. F3 double homozygote line was used for imaging.

Floral organs were removed with fine tweezers about 21 days after germination to expose inflorescence meristem. Meristems were mounted on the 60 mm plastic dish filled with 1.5% agar and imaged with a Zeiss LSM800 upright confocal microscope, equipped with a long working-distance water immersion objective ×60 (1 NA, Apochromat). Excitation was performed using a diode laser with 488 nm for GFP and 561 nm for TDT. Signal was collected at 500–550 and 600–660 nm, respectively. Images of three replicates were obtained.

### *Arabidopsis* root for PIN2 in 3D (*Figure 8N*, *Figure 8—figure supplement 2A*)

pPIN2::PIN2:GFP was previously described (*Xu and Scheres, 2005*). The seeds were stratified for 2 days at 4°C, grown on 1/2 Murashige and Skoog medium with 1% sucrose under the long-day condition (16 hr light/8 hr dark) at 20C°±2°C. The roots were stained by 10 µM propidium iodide (Sigma-Aldrich) and observed by Zeiss LSM780 with two-photon laser (excitation 990 nm) with a band pass filter 500/550 nm for GFP and 575–600 nm for PI. Images of three replicates were obtained.

## Data analysis

For the data analysis examples in the article, we computed all necessary cellular data within Morpho-GraphX and exported them as .csv files (see MorphoGraphX user guide Chapter 12.2 'Attribute maps' and Chapter 12.3 'Data export'). Those files were imported to RStudio for further processing or directly plotted using ggplot2 (*R Development Core Team, 2020*; *RStudio Team, 2019*; *Wickham, 2016*).

In the following, we provide a detailed description of the necessary processing steps for each data set shown in the figures. Hereby, we refer to the relevant chapters in the MorphoGraphX user guide which provides a step-by-step guide for most pipelines.

### *Arabidopsis* flower meristem (*Figure 1*)

We selected one sample for segmentation and further analysis. The segmentation, cell lineages, and heat maps were generated following the standard workflow as described in *Strauss et al., 2019* and in the MorphoGraphX user guide (Chapters 5–11).

### *Arabidopsis* ovule (*Figure 5*, *Figure 1—figure supplement 1*)

We selected one sample of the published data for segmentation and further analysis (*Vijayan et al., 2021*). Segmentation was obtained by back blending the raw images to CNN boundary predictions (*Wolny et al., 2020*) as described in *Vijayan et al., 2021*.

Following the segmentation (see also user guide Chapter 20 '3D segmentation'), for *Figure 1—figure supplement 1* we simply generated the heat map of the cell volumes (see user guide Chapter 21.1 'Heat maps & measures 3D').

For the additional analysis in *Figure 5*, we first generated a trimmed surface mesh (using the process 'Mesh/Creation/Marching Cubes Surface' on the segmented stack) and then used it to label the outermost layer using the process 'Mesh/Cell Atlas 3D/Ovule/Detect Cell Layers,' which is based

on the method described in *Montenegro-Johnson et al., 2019*. Next, we selected the cells of the cell-type (or parent) label of the outermost layer, inverted the selection, and deleted all other cells. At this stage, only the cells of the outer layer remained.

For each cell, its longitudinal organ axis (as custom cell axis X) was defined by a Bezier curve obtained from a manually selected central cell file using the processes 'Misc/Bezier/Bezier From Cell File' and 'Mesh/Cell Axis 3D/Custom/Create Bezier Line Directions' (see user guide Chapter 16.1 'Custom directions with Bezier' for general information about how to handle the Bezier in Morpho-GraphX and Chapter 21.6 'Custom directions for 3D meshes' for its application for organ coordinates). Next, the surface organ axis (as custom cell axis Y) was computed using the surface mesh and the process 'Mesh/Cell Axis 3D/Custom/Create Surface Direction.' Finally, the width direction (as custom cell axis Z) was computed as orthogonal direction of the other two using the process 'Mesh/Cell Axis 3D/Custom/Create Orthogonal Direction.'.

Next, the cell sizes were quantified by first doing a PCA on the voxels of cells in the segmented stack ('Mesh/Cell Axis 3D/Shape Analysis/Compute Shape Analysis 3D', see also user guide Chapter 22.2 'Cell shape analysis using principal component analysis for 3D meshes') and finally computing the component of the PCA's tensor aligned with the axes of interest ('Mesh/Cell Axis 3D/Shape Analysis/ Display Shape Axis 3D' with the appropriate 'Custom' heat option), with the Bezier direction corresponding to cell length, the surface direction to depth, and the orthogonal direction to width.

Shape anisotropy was defined using the equation: (max − 0.5*mid − 0.5*min)/(max + mid + min), with max, mid, and min defined by the length of the PCA axes. Elongation is defined by max/mid and flatness by mid/min.

To create the plots, cells with a distance >40 µm from the central cell file and few small cells at the distal end were filtered out. The data of the remaining 213 cells was plotted.

### *Arabidopsis* root (*Figure 2A*, *Figure 4B and D*, *Figure 4—figure supplement 1A and C*, *Figure 7A–D and I*, *Figure 7—figure supplement 1A–D*, *Figure 8—figure supplement 1A*)

From the three imaged replicates, we selected the sample with the best segmentation quality for further analysis. The raw images of the two time points of the analyzed root data sample were blurred and segmented using the ITK watershed segmentation processes in MorphoGraphX (see also *Stamm et al., 2017*) and the MorphoGraphX user guide Chapter 20 '3D segmentation'. From the segmented stack, a surface mesh and volumetric cell mesh were created using the processes 'Mesh/Creation/ Marching Cubes 3D' and 'Mesh/Creation/Marching Cubes Surface.'.

For the axis alignment analysis in *Figure 2A*, the organ was manually aligned with the y-axis and the coordinates of the cell centroids were computed (processes 'Mesh/Heat Map/Analysis/Cell Analysis 3D' and 'Mesh/Heat Map/Measures 3D/Location/Cell Coordinate' for the heat map; see also user guide Chapter 21.1 'Heat maps and measures 3D' and 22.3 'Further types of organ coordinates'). In *Figure 2B*, the cell volume of the 304 epidermis cells was plotted against the y-coordinate.

The 3D Cell Atlas pipeline (*Montenegro-Johnson et al., 2015*; *Stamm et al., 2017*) was used to compute cell coordinates, sizes, and cell types (*Figure 4B and D*, *Figure 4—figure supplement 1A–C*).

For the time-lapse analysis (*Figure 7A–I*, *Figure 7—figure supplement 1*), the cell lineages were determined semi-automatically using the pipeline introduced in this article (*Figure 6*) followed by a manual error correction (see also user guide Chapter 21.2 'Cell Lineage Tracking 3D'). Change maps were computed from the cells' volume and size data extracted from 3D Cell Atlas from both time points (see user guide Chapter 21.3 'Change maps 3D', Chapter 24.1 'Cell Atlas root' and *Stamm et al., 2017*). PDGs in 3D were derived from the deformation function mapping the first onto the second time point using parent-labeled cell centroids and cell wall centers (see user guide Chapter 21.4 'PDGs 3D').

For the analysis of the cell types in the endodermis (*Figure 7—figure supplement 1*), xylem cells in the stele and their neighboring pericycle cells were automatically identified by their circumferential coordinate obtained from the previously executed 3D Cell Atlas pipeline (see above). Endodermis cells touching two xylem-associated pericycle cells were determined as xylem pole. The two phloem poles in the endodermis were shifted by 90° (or two cells) from the xylem pole. In total, we used 26 xylem pole, 21 phloem pole, and 43 rest endodermis cells for the analysis.

For the analysis in *Figure 8E*, *Figure 8—figure supplement 1A*, only the second time point and its previously created parent labeling was used (see above). We computed the proliferation to the previous time point ('Mesh/Heat Map/Lineage Tracking/Heat Map Proliferation'), extracted the vertices on each division plane between cells that have divided exactly once (proliferation = 2, n = 249 mother cells that divided), and computed a PCA on each set of division plane vertices to extract the normals of the planes (using the process 'Mesh/Division Analysis/Analysis 3D/Division Analysis Multi,' see also user guide Chapter 25.2 'Division Analysis 3D'). Then, we computed the angle between the longitudinal axis of the organ as extracted by 3D Cell Atlas and the division planes ('Mesh/Division Analysis/Compute Division Plane Angles') and exported the data.

### *Arabidopsis* mature embryo (*Figure 2C*)

From more than 10 replicates, a sample with curved overall shape was selected for the demonstration of the organ coordinates using a Bezier curve. The fluorescence signal was segmented on the surface, and cells were parent labeled manually between two successive time points following the standard pipeline (see *Strauss et al., 2019*) and the MorphoGraphX user guide (Chapters 5–11). For creating the heat map of organ coordinates, a Bezier line was created and manually aligned with the organ (see user guide Chapter 23.2 'Bezier line and grid'). The organ coordinate heat map was then created using the process 'Mesh/Heat Map/Measures/Location/Bezier Line Coord.' The heat map is displayed on the later time point (after 48 hr of growth). Scale bars are displayed on the image.

### *Arabidopsis* sepal (*Figure 2D and E*, *Figure 2—figure supplement 1A and B*, *Figure 6D and E*, *Figure 6—figure supplement 1B*, *Figure 8A and D*)

For the sepal analysis, one replicate of the data from *Hervieux et al., 2016* consisting of seven time points was used (see *Figure 2—figure supplement 1A and B*).

For the analysis in *Figure 2D–F*, *Figure 2—figure supplement 1* for each time point, we manually determined the organ base based on the cell lineages from the first time point. Cells at the organ base were selected and used to compute the Euclidean cell distance measure ('Mesh/Heat Map/Measures/Location/Cell Distance,' see also the user guide Chapter 23.1 'The cell distance measure'). Finally, cell distances, growth, proliferation, and cell sizes were exported.

For the cell division analysis in *Figure 8A and B*, we analyzed the divisions that occurred between the time point T4 and T5. We computed the proliferation between these time points, extracted the vertices on each division plane between cells that have divided exactly once (proliferation = 2, n = 84), and computed a PCA on each set of division plane vertices to extract the normals of the planes (using the process 'Mesh/Division Analysis/Analysis 2D/Division Analysis Multi,' see user guide Chapter 25.1 'Division analysis 2.5D'). Next, we computed the PD-axis direction of the organ using the Euclidean cell distance from the base using the previously computed cell distance measure (see above) for creating custom directions along the PD-axis (see user guide Chapter 16.2 'Custom direction using a distance heat map'). Finally, we computed the angle between PD-axis and the division planes ('Mesh/Division Analysis/Compute Division Plane Angles') and exported the data.

For the growth analysis in *Figure 6D*, we computed the PDGs from time point T4 to time point T5, visualized on the earlier time point (see user guide Chapter 15 'Principal directions of growth (PDGs)'). *Figure 6E* shows the same time point, but here growth was computed using the gradient of the deformation function obtained from the cells' junctions (see user guide Chapter 17.6 'Growth analysis using deformation functions').

### *Arabidopsis* shoot apical meristem (*Figure 4A and C*, *Figure 4—figure supplement 1D*)

We selected one sample of the published data for analysis. Cel-type labels were determined using the methods described in 3D Cell Atlas meristem (*Montenegro-Johnson et al., 2019*). See also the user guide Chapter 24.2 'Cell Atlas meristem.' We computed the surface distance heat map using a surface mesh and the process 'Mesh/Heat Map/Measures 3D/Location/Mesh Distance' (see also user guide Chapter 23.3 'Further types of organ coordinates').

### *Arabidopsis* leaf (*Figure 3A–E*, *Figure 6—figure supplement 1A*, *Figure 8J and K*)

The *Arabidopsis* leaf data was previously published in *Kierzkowski et al., 2019*. One replicate of a time-lapse series consisting of seven time points was selected for analysis, but only time points T2 and T5 were used for the analysis here. The cell distance was computed similarly to the sepal example (*Figure 2D*) as distance from the organ base ('Mesh/Heat Map/Location/Cell Distance'; see also user guide Chapter 23.1 'The cell distance measure'). Additionally, we computed the heat map gradient of the cell distance heat map ('Mesh/Cell Axis/Custom/Create Heatmap Directions'; see user guide Chapter 16.2 'Custom direction using a distance heat map') to obtain custom directions along the PD axis and orthogonal to them along the ML axis of the organ for each cell. PDGs were computed and used to determine the amount of growth along the previously computed PD and ML axis (see user guide Chapter 15 'Principal directions of growth (PDGs)' and Chapter 16 'Custom axis directions').

For the morphing animation in *Figure 6—figure supplement 1A*, we used T2 and T5 and followed the user guide chapter 17.4 'Morphing animations'.

For the cell network analysis in *Figure 8J and K*, we computed the cell connectivity network of all cells in T5 weighted by the inverse of the length of the cell walls to determine the betweenness centrality. This is done by running the process 'Mesh/Heat Map/Measures/Network/Betweenness Centrality' (*Jackson et al., 2019*).

## Tomato shoot apical meristem (Figure 3F–I)

For the growth and DR5 signal analysis on the shoot apical meristem, we used one replicate of the previously published data of *Kierzkowski et al., 2012*. To objectively find the center of the meristem, primordium, and initiation site, the curvature of the cells was computed ('Mesh/Cell Axis/Curvature/Compute Tissue Curvature'). The resulting heat map was smoothed across neighboring cells for two rounds and resulting local maxima were identified as centers ('Mesh/Heat Map/Heat Map Smooth'). Meshes were manually aligned along the x-axis with respect to the meristem center to compute circumferential coordinates ('Mesh/Heat Map/Measures/Location/Polar Coord'; see also user guide Chapter 23.3 'Further types of organ coordinates') around the primordium and initiation center. For the analysis, only cells in the vicinity of the primordium and initiation centers were considered (as obtained by the cell distance towards their center cell using 'Mesh/Heat Map/Measures/Location/Cell Distance'; see also user guide Chapter 23.1 'The cell distance measure'). Furthermore, the gradients of the Euclidean cell distance heat maps from both centers were used to compute custom directions along the heat (=radial) and orthogonal to the heat (=circumferential) (using the process 'Mesh/Cell Axis/Custom/Create Heatmap Directions'; see also user guide Chapter 16.2 'Custom directions using a distance heat map'). Finally, the growth analysis was done similarly to the leaf, computing PDGs and growth along the custom axis ('Mesh/Cell Axis/PDG/Compute Growth Directions'; see also user guide Chapter 15 'Principal directions of growth (PDGs)' and Chapter 16 'Custom axis directions').

After each step, heat maps were exported to attribute maps and in the end exported to .csv files (see also Chapter 12 'Attribute maps & data export').

### *Arabidopsis* gynoecium (*Figure 4E–H*)

We selected one replicate for further analysis. After the surface segmentation (see user guide Chapters 5–9), we computed the heat map for the length of the minor axis in *Figure 4E* (process 'Mesh/Heat Map/Measures/Geometry/Length Minor Axis'; see also user guide Chapter 10 'Cell geometry quantification' and Chapter 22 'Cell shape analysis using PCA'). For *Figure 4F*, after a manual alignment of the mesh we computed the x-coordinate of the cells (process 'Mesh/Heat Map/Measures/Location/Cell Coordinate'; see also user guide Chapter 23.3 'Further types of organ coordinates'). For both of these heat maps, cell types were generated by determining an appropriate threshold and selecting cells by their heat value using the process 'Mesh/Heat Map/Heat Map Select' to set their parent label (=cell-type label). For more details, see also user guide Chapter 24.4 'Cell type classification using a single heat map.'.

In *Figure 4G*, we created a clustering using the process 'Mesh/Cell Types/Classification/Tools/Cell Property Map 2D.' See also the user guide Chapter 24.6 'Cell type classification using two measures.'.

In *Figure 4H*, we used the SVM training and classification pipeline to generate the cell-type labels from a small training set (as shown in the figure). See also the user guide Chapter 24.7 'Cell type classification using SVMs.'.

### *Arabidopsis* leaf (*Figure 6B and C*)

We selected one replicate for segmentation, parent labeling, and the demonstration of the semi-automatic parent labeling. See user guide Chapters 5–9 about creating a surface segmentation and Chapter 17.2 'Semi-automatic parent labeling' for more details.

### *Arabidopsis* shoot apical meristems (*Figure 8L and M*, *Figure 8—figure supplement 2E and F*)

For the MT analysis, we selected one sample for segmentation and analysis. We determined the center of organ based on a smoothed curvature heat map ('Mesh/Cell Axis/Curvature/Compute Tissue Curvature'). The center cell was selected and the Euclidean cell distance to the remaining cells was computed ('Mesh/Heat Map/Location/Cell Distance'). The circumferential direction around the cell center was obtained from the orthogonal direction of the heat map directions ('Mesh/Cell Axis/Custom/Create Heatmap Directions'). Cells were then binned by their Euclidean distance to the center ('Mesh/Heat Map/Operators/Heat Map Binning').

### *Arabidopsis* embryo (*Figure 8F and G*, *Figure 8—figure supplement 1B–E*)

The data for the 3D division analysis in *A. thaliana* embryos was previously published in *Yoshida et al., 2014*. From this dataset, we chose one wildtype and one inducible bdl (pRPS5a>>bdl) sample at the 16-cell stage.

A surface mesh was generated from the cells in the embryo, and the cells were parent labeled according to their predicted mother cell. Then, the process 'Mesh/Division Analysis/Analysis 3D/Division Analysis Multi' performed the following steps on all of the parent-labeled cells (n = 16 cells or 8 divisions in each genotype): first, a planar approximation of the actual division plane was computed by performing a PCA on the vertex positions of the shared wall between the two daughter cells. Then, 1000 equally distributed division planes were simulated on the combined mother cell and different measures were quantified. See also the user guide Chapter 25.2 'Division analysis 3D' for more details. The actual and the best planes were visualized using the process 'Mesh/Division Analysis/Display and Filter Planes.'.

### *Arabidopsis* root PIN2 in 3D (*Figure 8N*, *Figure 8—figure supplement 2A and B*)

For the analysis of the PIN directions in the *A. thaliana* root, we selected one sample for 3D segmentation (see user guide Chapter 20 '3D segmentation'). Next, we defined the main organ axis using a Bezier curve through the center of the organ ('Mesh/Cell Axis 3D/Custom/Create Bezier Line Directions'; see also user guide Chapter 21.6 'Custom directions for 3D meshes'). Then, we computed the PIN2 polarity direction ('Mesh/Cell Axis 3D/Polarization/Compute Signal Orientation'; see also user guide Chapter 21.7 'Signal orientation for 3D meshes') and determined the angle between the polarity direction and the Bezier line ('Mesh/Cell Axis 3D/Compute Angles').

## Acknowledgements

We acknowledge support by the Center for Advanced Light Microscopy of the Technical University of Munich School of Life Sciences and inspiration from the many collaborators we have worked with during the development of MorphoGraphX. We thank Jim Haseloff for sharing transgenic marchantia lines. We would also like to thank the reviewers and the editor for their helpful comments and suggestions.

## Additional information

### Funding

| Funder | Grant reference number | Author |
|---|---|---|
| Deutsche Forschungsgemeinschaft | Forschunggruppe 2581 | Kay Schneitz<br>Miltos Tsiantis<br>Richard S Smith |
| Biotechnology and Biological Sciences Research Council | ISP to John Innes Centre (BB/P013511/1) | Richard S Smith |
| Bundesministerium für Bildung und Forschung | 031A494 & 031A492 | Richard S Smith |
| Deutsche Forschungsgemeinschaft | STE2802/2-1 | Dennis Eschweiler |
| New Frontiers in Research Fund | 2018-00953 | Anne-Lise Routier-Kierzkowska<br>Daniel Kierzkowski |
| Natural Sciences and Engineering Research Council of Canada | RGPIN-2018-04897 | Daniel Kierzkowski |
| Natural Sciences and Engineering Research Council of Canada | RGPIN-2018-05762 | Anne-Lise Routier-Kierzkowska |
| Leverhulme Trust | RPG-2019-267 | George W Bassel |
| Biotechnology and Biological Sciences Research Council | BB/S002804/1 | George W Bassel |
| Human Frontier Science Program | RGP0002/2020 | George W Bassel |
| Max Planck Society | Core grant | Miltos Tsiantis |
| Fonds de Recherche du Québec Nature et Technologies | 282285 | Anne-Lise Routier-Kierzkowska<br>Daniel Kierzkowski |
| Deutsche Forschungsgemeinschaft | ERA-CAPS V-Morph | Richard S Smith |

The funders had no role in study design, data collection and interpretation, or the decision to submit the work for publication.

### Author contributions

Sören Strauss, Data curation, Formal analysis, Investigation, Methodology, Software, Validation, Visualization, Writing - original draft, Writing – review and editing; Adam Runions, Data curation, Formal analysis, Investigation, Methodology, Resources, Software, Validation, Visualization, Writing – review and editing; Brendan Lane, Formal analysis, Investigation, Methodology, Resources, Software, Validation, Visualization, Writing – review and editing; Dennis Eschweiler, Johannes Stegmaier, Software, Writing – review and editing; Namrata Bajpai, Formal analysis, Investigation, Methodology, Software, Validation, Visualization, Writing – review and editing; Nicola Trozzi, Data curation, Investigation, Resources, Validation, Writing – review and editing; Anne-Lise Routier-Kierzkowska, Investigation, Methodology, Software, Validation, Visualization, Writing – review and editing; Saiko Yoshida, Athul Vijayan, Data curation, Formal analysis, Investigation, Writing – review and editing; Sylvia Rodrigues da Silveira, Rachele Tofanelli, Emillie Echevin, Constance Le Gloanec, Hana Bertrand-Rakusova, Investigation, Writing – review and editing; Mateusz Majda, Data curation, Investigation, Validation, Writing – review and editing; Milad Adibi, Data curation, Validation, Writing – review and editing; Kay Schneitz, Funding acquisition, Supervision, Writing – review and editing; George W Bassel, Formal analysis, Investigation, Supervision, Validation, Writing – review and editing; Daniel Kierzkowski,

Investigation, Supervision, Writing – review and editing; Miltos Tsiantis, Conceptualization, Funding acquisition, Resources, Supervision, Writing – review and editing; Richard S Smith, Conceptualization, Data curation, Formal analysis, Funding acquisition, Investigation, Methodology, Project administration, Resources, Software, Supervision, Validation, Visualization, Writing - original draft, Writing – review and editing

### Author ORCIDs
Sören Strauss http://orcid.org/0000-0002-7141-1750
Adam Runions http://orcid.org/0000-0002-7758-7423
Nicola Trozzi http://orcid.org/0000-0003-3951-6533
Anne-Lise Routier-Kierzkowska http://orcid.org/0000-0003-0383-0811
Athul Vijayan http://orcid.org/0000-0003-1837-6359
Rachele Tofanelli http://orcid.org/0000-0002-5196-1122
Mateusz Majda http://orcid.org/0000-0003-3405-2901
Constance Le Gloanec http://orcid.org/0000-0002-7959-6307
Kay Schneitz http://orcid.org/0000-0001-6688-0539
Daniel Kierzkowski http://orcid.org/0000-0002-1947-8691
Johannes Stegmaier http://orcid.org/0000-0003-4072-3759
Richard S Smith http://orcid.org/0000-0001-9220-0787

### Decision letter and Author response
Decision letter https://doi.org/10.7554/eLife.72601.sa1
Author response https://doi.org/10.7554/eLife.72601.sa2

---

## Additional files

### Supplementary files
• Transparent reporting form

### Data availability
All new data collected for this article, along with the specific version of MorphoGraphX (https://www.MorphoGraphX.org) used for analysis, and the tutorial videos are available online, and can be downloaded from Dryad at: https://doi.org/10.5061/dryad.m905qfv1r.

The following dataset was generated:

| Author(s) | Year | Dataset title | Dataset URL | Database and Identifier |
|---|---|---|---|---|
| Smith RS | 2022 | MorphoGraphX2: Datasets that demonstrate how to create positional information with local coordinate systems | https://doi.org/10.5061/dryad.m905qfv1r | Dryad Digital Repository, 10.5061/dryad.m905qfv1r |

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
