## [Editor Report]

Quantitative imaging has become a mainstay of modern cell and developmental biology. This article reports major advances in the image analysis software package MorphGraphX (MGX). MGX2.0 includes new tools for precise quantitation of cellular behaviors, such as cell division and expansion, within the context of positional information in the growing organs. This article is the go-to resource for current and future users of MGX to learn the power of the software package, with which they can quantify the spatiotemporal dynamics of the growth and development of living organisms.

---

## [Decision Letter]

**Decision letter after peer review:**

Thank you for submitting your article "MorphoGraphX 2.0: Providing context for biological image analysis with positional information" for consideration by *eLife*. Your article has been reviewed by 3 peer reviewers, one of whom is a member of our Board of Reviewing Editors, and the evaluation has been overseen by Detlef Weigel as the Senior Editor. The following individuals involved in review of your submission have agreed to reveal their identity: Roeland MH Merks (Reviewer #2); Naomi Nakayama (Reviewer #3).

Essential revisions:

1) All reviewers agree that MGX2.0 is a powerful suite of tools that can be of use to many researchers in the plant and developmental biology fields, but all felt it is more appropriate for a "Tools and Resources" manuscript and suggest that the revision be submitted as such.

2) Even for a tools and resources manuscript, it would be nice to clarify what novel findings MorphGraphX can reveal with the new updates. Did you find something counterintuitive or previously unknown/unexpected because of the precision quantification? Did using these data result in different outcomes with a computational model, for example? Such an example will be a strong incentive for why these updates warrant another major publication.

3) More detail on how the data must be generated to make it suitable for use in MGX2.0, such as explicit parameters for input image quality, is required. MGX1.0 had fairly strict standards for input. Have these standards changed? Also, in the last 10 years, many post-acquisition programs exist to "clean-up" noisy image data (e.g. ilastik workflows). Are these packages compatible with MGX2.0?

4) Provide more clearly organized workflow overviews. It is impressive how many analysis tools are available, but so much information is overwhelming. Even as experienced users of MGX1.0, (Rev 1), it was challenging to follow what tools were being applied to what questions and tissues. To reach both beginner and expert MGX users, we suggest including a table that summarizes the approaches and applications described in the paper.

5) To help get users started with MGX2.0, (1) present the software availability more prominently along the software availability sections; (2) add to each figure a set of instructions/a protocol for reproducing the figure using MGX2.0 as part of the supplements. At least for one of these a highly detailed step-by-step set of instructions would be highly useful; the next figures could build upon these.

6) The Results section is challenging to read and needs some restructuring. Specifically (1) the figures are described out of order, and most figures are scattered in different main text sections. Consolidate the order of the information according to the messages of each figure or rearrange figure panels; (2) The basic procedure of the software use is introduced in the middle (Line 360-370), but this should be in the introduction or beginning of the results; (3) multiple organ and tissue types are juxtaposed in the examples, of making the text difficult to follow. Some context and background will be needed to flesh out why it's important to quantify specific morphogenetic events in the mutants or markers chosen as examples.

7) Please clarify the novelty of this report beyond the other publications on MorphGraphX in the past five years (e.g., Montenegro-Johnson et al., 2015; Montenegro-Johnson et al., 2019; Strauss et al., 2019).

Specific examples of these general points about organization and narrative are found in the three reviews. Please consider these specifics as you address the required changes noted in #1-#7.

*Reviewer #1 (Recommendations for the authors):*

Items that must be addressed

– Image quality required. It is unclear whether there are any changes in regard to the input data (image quality/step size) required for the processes described. Even if there are no changes it is unclear for those new to using MorphographX and could lead to disappointment. A short paragraph on input data would be of great help.

– Tomato meristem. The paragraph about auxin in tomato meristems (line 299-308) is confusing and could do with some clarification. The conclusion is too strong for the data presented. The authors should either rephrase OR present experiments in which manipulations took place OR cite previous work that adds to the presented data.

– Workflow overview. The paper provides a great overview of new workflows introduced in MorphographX 2.0. Because there is so much information, it can be overwhelming. We suggest including a table that summarizes the approaches and applications described in the paper. This would allow beginner and expert users to see all that is on offer in an efficient way.

Items recommended to take into consideration

– Intended audience. We are assuming the intended audience are plant biologists that either have experience using MorphographX or are interested in using the software in the future. With that assumption it would be useful to clarify some terms as they might be unclear for readers. Such as Bezier splines/surface. In addition other sections provide detail that is confusing rather than clarifying (eg line 775-795 will cause many readers to get lost just before reaching the end of the story).

– Central story. One of the strengths of the story is the many different tissues used in the story. However, this also sometimes hinders the coherence/flow. We realize this is probably not realistic for this story but for future stories the authors could consider picking one central story to revisit at the start of each section before diving into other tissues/applications.

– Segmentation. There are many new functions in MorphographX 2.0, how did any previously existing applications change? Most importantly, did the segmentation/merging change at all? Manual correction of segmentation errors is the most time-consuming step for many casual users.

– Figures/story structure. Figures and story don't use the same story structure which gets confusing in the first half of the story. The story only partially discusses figures at first and then gets back to them with one section moving between figures 2, 5 and 4 and the next section discussing parts of figure 3. For the reader this is confusing as (1) when first encountering the figure they might try to understand the figure as a whole, thus viewing all the panels while only several are discussed and then (2) they are later directed to jump between figures+pages to follow the story which is quite impractical. There is no simple solution, but we encourage the authors to consider either restructuring some earlier figures.

– Plot axes. Putting the dependent variable on the x-axis of many of the plots is visually pleasing and matches the orientation of many samples. But it can be confusing to many readers who are used to seeing the dependent variable on the y-axis.

– Advanced geometric analysis. This section if the least coherent and most difficult to follow. Obviously, there are analyses that cannot have a whole section that should be discussed. A short introductory sentence in this section might help make that clear so readers are not looking for logic that is not there. In addition, some of the paragraphs should be shortened or rewritten. They are currently both long and confusing.

– Exploded views/3D visualization. The paragraph describing 3D visualization (line 683-699) describes difficulties with 3D visualization and possible solutions. From the text it's not entirely clear how this works in MorphographX 2.0. Can certain bundles of cells be unselected/made invisible when navigator/exploring segmentation? What are the limits of these approaches? Exploded views might work great for young embryos but less well for larger tissues.

– Comparison to comparable software. MorphographX 2.0 is a great tool. It would be great to have its applications/strengths (discussed in line 762 onwards) put into context compared to other similar tools (Ilastik, etc).

Additional questions we had about the authors visions. These do not need to be addressed for the paper to be meaningful, but they were questions that immediately came to mind when reading this. If the authors want to add to the nascent sections describing future applications, below are some things that may be of interest to a plant development audience.

– Cells as nodes in networks. This is touched upon and sounds interesting. Do the authors imaging visualizing cell networks in cytoscape? In addition to the one story mentioned, do you imagine other applications?

– Positional + gene expression data. The importance of positional information in addition to other large datasets such as expression data is mentioned several times. How do the authors imagine that this information could be integrated in the future?

*Reviewer #2 (Recommendations for the authors):*

To help get users started, it would be helpful to (1) present the software availability more prominently along the software availability sections; (2) add to each figure a set of instructions/a protocol for reproducing the figure using MorphoGraphX, e..g as part of the supplements. At least for one of these a highly detailed step-by-step set of instructions would be highly useful; the next figures could build upon these.

I am a Mac user and have not been able to download and try the software yet, so perhaps instructions are in the source code package.. However,. I tried clicking the "Get Help" link on the MorphoGraphX main page, but the link is broken. In any case it would help if it is more clearly communicated to readers that they can do all of these cool analyses themselves, and give them instructions or directions to step-by-step instructions for performing the analyses themselves.

*Reviewer #3 (Recommendations for the authors):*

It is a manuscript with beautiful visual figures, reporting a widely useful open-source image analysis tool. However, I strongly recommend that further improvement before publication especially around the below points.

1) I found the Results section confusing – restructuring would be helpful.

– The figures are described out of order, and most figures are scattered in different main text sections. Can you consolidate the order of the information according to the messages of each figure? Perhaps the figures should be rearranged?

– The basic procedure of the software use is introduced in the middle (e.g., information in Line 360-370 should be in the introduction or at the beginning of the results?).

– The Results section illustrates multiple types of organs in a juxtaposition. This makes it a little difficult to follow. At least include enough information (e.g., what is bdl, DR5, etc.) for the particular cases of the development of these organs so that the readers can appreciate why it is important to quantitate specific morphogenetic events in these mutants or with these markers.

2) What is the article type? The manuscript does not seem to report a major new finding of a biological phenomenon; hence, it is more suitable as a tools and resources paper rather than a research article?

3) Even for a tools and resources manuscript, it would be nice to clarify what novel findings MorphGraphX can reveal with the new updates. Did you find something counterintuitive or previously unknown/unexpected because of the precision quantification? Did using these data result in different outcomes with a computational model, for example? Such an example will be a strong incentive for why these updates warrant another major publication.

4) Please clarify the novelty of this report beyond the other publications on MorphGraphX in the past five years (e.g., Montenegro-Johnson et al., 2015; Montenegro-Johnson et al., 2019; Strauss et al., 2019).

---

## [Author Response]

Essential revisions:1) All reviewers agree that MGX2.0 is a powerful suite of tools that can be of use to many researchers in the plant and developmental biology fields, but all felt it is more appropriate for a "Tools and Resources" manuscript and suggest that the revision be submitted as such.

Yes, it should have been designated as a tools and resources manuscript. We had assumed that since it was submitted as a “Research Advance” on a previous “Tools and Resources” manuscript, that it would inherit the same classification. In any case, please change it to a "Tools and Resource" manuscript, and if it is possible to indicate is as an advance on the original MGX manuscript that would also be nice.

2) Even for a tools and resources manuscript, it would be nice to clarify what novel findings MorphGraphX can reveal with the new updates. Did you find something counterintuitive or previously unknown/unexpected because of the precision quantification? Did using these data result in different outcomes with a computational model, for example? Such an example will be a strong incentive for why these updates warrant another major publication.

In the initial submission we had mostly focused on the methods themselves, so we have tried to be more explicit about the results that can be had with the new tools that were not possible/difficult before.

3) More detail on how the data must be generated to make it suitable for use in MGX2.0, such as explicit parameters for input image quality, is required. MGX1.0 had fairly strict standards for input. Have these standards changed? Also, in the last 10 years, many post-acquisition programs exist to "clean-up" noisy image data (e.g. ilastik workflows). Are these packages compatible with MGX2.0?

The requirements for high quality datasets are largely the same, and the guidelines from MGX1.0 are still in the user guide, with a few small updates. Some datasets can be improved using the CNN methods, so we have added discussion of this. We have also emphasized the point, that although high quality data sets are still required, especially when attempting a full 3D segmentation and analysis, surface (2.5D) processing can be done on much lower quality samples. The coordinate systems in some cases can also lower this threshold a bit more, for example on roots or hypocotyls where one may only be interested in cell length, and then the segmentation does not have to be as accurate. We have added these points to the text in an introductory section at the beginning of the results and added to the user guide.

4) Provide more clearly organized workflow overviews. It is impressive how many analysis tools are available, but so much information is overwhelming. Even as experienced users of MGX1.0, (Rev 1), it was challenging to follow what tools were being applied to what questions and tissues. To reach both beginner and expert MGX users, we suggest including a table that summarizes the approaches and applications described in the paper.

It has indeed become a rather large and complicated software, and we have tried to make things as accessible as possible for new users, while still explaining the advanced features for more experienced readers. In addition to the short videos provided in the initial submission, we have added longer tutorial-style videos for some of the more involved analysis, such as the creating of coordinate systems, semi-automatic lineage tracking, cell type classification, deformation functions for morphing and animating growth, and use of the deep learning tools. As they are very large, we have included them in the data repository for the manuscript, and have referenced them in the figures. We have also added a table which gives an overview of the available workflows, also highlighting what is new, and with references to the figures and where the workflows are used, and the chapters in the user guide that contain the step by step guides. Finally, we have extended the methods section to give a more detailed description on what processes and pipelines were used to achieve the results of the different analyses shown in the figures. We hope that we have now provided enough resources for the range of users from beginners to experts, to enable the repetition of all the analysis pipelines shown, either with the data we have provided, and with their own data sets.

5) To help get users started with MGX2.0, (1) present the software availability more prominently along the software availability sections; (2) add to each figure a set of instructions/a protocol for reproducing the figure using MGX2.0 as part of the supplements. At least for one of these a highly detailed step-by-step set of instructions would be highly useful; the next figures could build upon these.

As requested, we have made the presentation more prominent in the software availability section. There already was a loose connection between a lot of the figures and the user guide, as we have tried to use the same samples, which are also available to download. We agree that this was not very obvious so we have now put references in the figure captions. We have also extended the methods section to provide more detail for each figure and its relevant sections of the user guide. As mentioned in (4), there are now extended video tutorials for some of the more advanced new features that provide click-by-click detail (and sound) on exactly what to do using the sample data sets.

6) The Results section is challenging to read and needs some restructuring. Specifically (1) the figures are described out of order, and most figures are scattered in different main text sections. Consolidate the order of the information according to the messages of each figure or rearrange figure panels; (2) The basic procedure of the software use is introduced in the middle (Line 360-370), but this should be in the introduction or beginning of the results; (3) multiple organ and tissue types are juxtaposed in the examples, of making the text difficult to follow. Some context and background will be needed to flesh out why it's important to quantify specific morphogenetic events in the mutants or markers chosen as examples.

We have restructured the text as suggested, changing the figure order and text flow to better match the figures. We have moved some of the text from lines 360-370 to the beginning of the results. As mentioned, some figures show multiple organs to demonstrate the application of organ systems (Figure 2) or cell type classification (Figure 4) or of the additional tools (Figure 8), so in these cases we have added more context to the samples chosen to illustrate the utility of the various tools presented.

7) Please clarify the novelty of this report beyond the other publications on MorphGraphX in the past five years (e.g., Montenegro-Johnson et al., 2015; Montenegro-Johnson et al., 2019; Strauss et al., 2019).

We have added to the discussion to explain the relationship of MorphoGraphX to these previous works.

Specific examples of these general points about organization and narrative are found in the three reviews. Please consider these specifics as you address the required changes noted in #1-#7.Reviewer #1 (Recommendations for the authors):Items that must be addressed– Image quality required. It is unclear whether there are any changes in regard to the input data (image quality/step size) required for the processes described. Even if there are no changes it is unclear for those new to using MorphographX and could lead to disappointment. A short paragraph on input data would be of great help.

We have added this, see also response to the editor summary comment 3.

– Tomato meristem. The paragraph about auxin in tomato meristems (line 299-308) is confusing and could do with some clarification. The conclusion is too strong for the data presented. The authors should either rephrase OR present experiments in which manipulations took place OR cite previous work that adds to the presented data.

We have tried to clarify the text, toned down the conclusion, and added references for the notion that auxin acts a trigger for primordium initiation.

– Workflow overview. The paper provides a great overview of new workflows introduced in MorphographX 2.0. Because there is so much information, it can be overwhelming. We suggest including a table that summarizes the approaches and applications described in the paper. This would allow beginner and expert users to see all that is on offer in an efficient way.

As suggested, we have added a table of workflows with references to the related chapters in the user guide (see also response to the editor summary comment 4).

Items recommended to take into consideration– Intended audience. We are assuming the intended audience are plant biologists that either have experience using MorphographX or are interested in using the software in the future. With that assumption it would be useful to clarify some terms as they might be unclear for readers. Such as Bezier splines/surface. In addition other sections provide detail that is confusing rather than clarifying (eg line 775-795 will cause many readers to get lost just before reaching the end of the story).

The paper is intended for a dual audience, biologists that use the tools, as well as computational people that might be interested in using similar methods in their work, or even contributing to MGX. We have moved the paragraph indicated by the reviewer to its own section on software design, and have clarified any terms used and provided more background for the aspects covered there. The intent in this section was to give advanced users and computational people a brief overview of the many new features that have been integrated into MorphoGraphX, and hopefully encourage others to join in further development.

– Central story. One of the strengths of the story is the many different tissues used in the story. However, this also sometimes hinders the coherence/flow. We realize this is probably not realistic for this story but for future stories the authors could consider picking one central story to revisit at the start of each section before diving into other tissues/applications.

As mentioned above, we decided to follow the development of coordinate systems as a central story, however this makes it necessary to introduce many biological “mini-stories” as the development of coordinate systems and their application does not map nicely to a single biological story. Nevertheless, we have tried to smooth this out in the revised version (see also editor summary answer 4).

– Segmentation. There are many new functions in MorphographX 2.0, how did any previously existing applications change? Most importantly, did the segmentation/merging change at all? Manual correction of segmentation errors is the most time-consuming step for many casual users.

The segmentation itself is still using watershed for 2.5D (our custom code) or 3D (ITK libraries). What has made the biggest difference is pre- and post-processing methods, as well as tools for correction. One substantial improvement is the ability to do semi-automatic lineage tracking using the deformation functions that can make a continuous map from one time point to another. By labeling a few cells, the map can be used to fill in the rest and then they are verified based on neighborhood relations. This is probably a whole method in itself. Another substantial improvement comes with pre-processing using CNN (convolutional Neural Networks), that can dramatically improve weak signal, especially for 3D work where the quality needs to be much higher for 3D. MGX can also readily load processed image or segmented data that have been processed on other platforms. We have added an overview of these possibilities to an outline paragraph at the beginning of the results.

– Figures/story structure. Figures and story don't use the same story structure which gets confusing in the first half of the story. The story only partially discusses figures at first and then gets back to them with one section moving between figures 2, 5 and 4 and the next section discussing parts of figure 3. For the reader this is confusing as (1) when first encountering the figure they might try to understand the figure as a whole, thus viewing all the panels while only several are discussed and then (2) they are later directed to jump between figures+pages to follow the story which is quite impractical. There is no simple solution, but we encourage the authors to consider either restructuring some earlier figures.

We have tried to clean this up by rearranging the figures and the text (see also editor summary answer 4 and 6).

– Plot axes. Putting the dependent variable on the x-axis of many of the plots is visually pleasing and matches the orientation of many samples. But it can be confusing to many readers who are used to seeing the dependent variable on the y-axis.

We have changed to orientation of the samples and the axes (see Figure 2, 2S1, 4, 7, 7S1).

– Advanced geometric analysis. This section if the least coherent and most difficult to follow. Obviously, there are analyses that cannot have a whole section that should be discussed. A short introductory sentence in this section might help make that clear so readers are not looking for logic that is not there. In addition, some of the paragraphs should be shortened or rewritten. They are currently both long and confusing.

This section was a catch-all for things we thought too important not to be mentioned, but didn’t fit anywhere else. To minimize confusion, we have organized it into short sub-sections, so that each will be seen as somewhat stand alone. We have also done as the reviewer suggests and added an introductory sentence, and tried to streamline and shorten the text here as much as possible.

– Exploded views/3D visualization. The paragraph describing 3D visualization (line 683-699) describes difficulties with 3D visualization and possible solutions. From the text it's not entirely clear how this works in MorphographX 2.0. Can certain bundles of cells be unselected/made invisible when navigator/exploring segmentation? What are the limits of these approaches? Exploded views might work great for young embryos but less well for larger tissues.

Cells cannot be hidden from view by selecting (other than deleting them), although we are constantly improving the visualization tools, and this would be a nice option to have. Since the exploded view can be used with multiple clipping planes, it does a lot better on larger samples than one would initially expect. We have been more specific in the revised version about what this feature can do.

– Comparison to comparable software. MorphographX 2.0 is a great tool. It would be great to have its applications/strengths (discussed in line 762 onwards) put into context compared to other similar tools (Ilastik, etc).

We have added a discussion of how MorphoGraphX relates to other software in the Software design section.

Additional questions we had about the authors visions. These do not need to be addressed for the paper to be meaningful, but they were questions that immediately came to mind when reading this. If the authors want to add to the nascent sections describing future applications, below are some things that may be of interest to a plant development audience.– Cells as nodes in networks. This is touched upon and sounds interesting. Do the authors imaging visualizing cell networks in cytoscape? In addition to the one story mentioned, do you imagine other applications?

There are a few measures that treat the cells as nodes in a graph, with the weights of the cell-cell interfaces determined by area, but of course very limited in comparison to what Cytoscape can do. It is possible to export the network information to.csv files, which can then be loaded into Cytoscape.

– Positional + gene expression data. The importance of positional information in addition to other large datasets such as expression data is mentioned several times. How do the authors imagine that this information could be integrated in the future?

Probably the biggest obstacle here is the lack of cell level gene expression information. Nevertheless some have already started to combine gene expression data and present it visually on template organs, for example the Plant Cell Atlas group. For this type of data, it is straightforward to write plugins to import it and visualize on cellular geometry, now matter what software it was generated in. MGX can already read a wide array of mesh, and voxel data in different formats, including some metadata.

Reviewer #2 (Recommendations for the authors):To help get users started, it would be helpful to (1) present the software availability more prominently along the software availability sections; (2) add to each figure a set of instructions/a protocol for reproducing the figure using MorphoGraphX, e..g as part of the supplements. At least for one of these a highly detailed step-by-step set of instructions would be highly useful; the next figures could build upon these.

As mentioned above (this comment was highlighted in the editor summary answers 4 and 5), we have made the presentation of the software availability more prominent. We have also added more description in the methods and the user guide and videos to address this point.

I am a Mac user and have not been able to download and try the software yet, so perhaps instructions are in the source code package.. However,. I tried clicking the "Get Help" link on the MorphoGraphX main page, but the link is broken. In any case it would help if it is more clearly communicated to readers that they can do all of these cool analyses themselves, and give them instructions or directions to step-by-step instructions for performing the analyses themselves.

Sadly there is no Mac version, although the it does run on Parallels and VMWare (recommend the non-Cuda Linux version). It did run on Mac at one time. The problem is that Mac has been removing OpenGL support over the years, so it is not possible to create an OpenGL context that supports all the features needed in MGX. Soon the Mac will drop OpenGL entirely, so we would need to do a port to Metal. Not impossible, but we are currently lacking resources for that.

The link should now be fixed, it just points to the Help page of the site.

We have now added some very detailed step-by-step videos that use the datasets provided with the manuscript, with audio and click-by-click instructions.

Reviewer #3 (Recommendations for the authors):It is a manuscript with beautiful visual figures, reporting a widely useful open-source image analysis tool. However, I strongly recommend that further improvement before publication especially around the below points.1) I found the Results section confusing – restructuring would be helpful.– The figures are described out of order, and most figures are scattered in different main text sections. Can you consolidate the order of the information according to the messages of each figure? Perhaps the figures should be rearranged?

We have done as the reviewer suggested (see also the editor summary answer 6).

– The basic procedure of the software use is introduced in the middle (e.g., information in Line 360-370 should be in the introduction or at the beginning of the results?).

We have moved this to an introductory section added at the beginning of the results (see also the editor summary answer 6).

– The Results section illustrates multiple types of organs in a juxtaposition. This makes it a little difficult to follow. At least include enough information (e.g., what is bdl, DR5, etc.) for the particular cases of the development of these organs so that the readers can appreciate why it is important to quantitate specific morphogenetic events in these mutants or with these markers.

We have clarified terms where they were missing, and added more context to the biological results that are used as examples.

2) What is the article type? The manuscript does not seem to report a major new finding of a biological phenomenon; hence, it is more suitable as a tools and resources paper rather than a research article?

I think this was a misunderstanding. It was submitted as a “Research Advance” on the MGX 1 paper, with the hopes they would be linked together on the *eLife* site, so we assumed it would inherit the “Tools and Resources” designation from there. In any case, the reviewer is correct, it should be tools and resources (see also the editor summary answers 1 and 2).

3) Even for a tools and resources manuscript, it would be nice to clarify what novel findings MorphGraphX can reveal with the new updates. Did you find something counterintuitive or previously unknown/unexpected because of the precision quantification? Did using these data result in different outcomes with a computational model, for example? Such an example will be a strong incentive for why these updates warrant another major publication.

We have added more biological context to the examples we use to illustrate the tools, and more discussion of the relevance of the findings, and in particular aspects that would not be possible without the new tools.

4) Please clarify the novelty of this report beyond the other publications on MorphGraphX in the past five years (e.g., Montenegro-Johnson et al., 2015; Montenegro-Johnson et al., 2019; Strauss et al., 2019).

We have a added a discussion of the relationship between MGX 2.0 and our previous work mentioned by the reviewer in the conclusions.